# How is Baseflow Index (BFI) impacted by water resource management practices?

John P. Bloomfield[1], Mengyi Gong[2,3], Benjamin P. Marchant[1], Gemma Coxon[4], Nans Addor[5]

[1]British Geological Survey, Wallingford, OX10 8BB, UK
[2]British Geological Survey, Keyworth, NG12 5GG, UK
[3]Lancaster University, Lancaster, LA1 4YF, UK
[4]University of Bristol, Bristol, BS8 1SS, UK
[5]University of Exeter, Exeter, EX4 4RJ, UK

*Correspondence to*: John P. Bloomfield (jpb@bgs.ac.uk)

**Abstract.** Water resource management (WRM) practices, such as groundwater and surface water abstractions and effluent discharges, may impact baseflow. Here the CAMELS-GB large-sample hydrology dataset is used to assess the impacts of such practices on baseflow index (BFI) using statistical models of 429 catchments from Great Britain. Two complementary modelling schemes, multiple linear regression (LR) and machine learning (random forests, RF), are used to investigate the relationship between BFI and two sets of covariates (natural covariates only and a combined set of natural and WRM covariates). The LR and RF models show good agreement between explanatory covariates. In all models, the extent of fractured aquifers, clay soils, non-aquifers, and crop cover in catchments, catchment topography and aridity are significant or important natural covariates in explaining BFI. When WRM terms are included, groundwater abstraction is significant or the most important WRM covariate in both modelling schemes and effluent discharge to rivers is also identified as significant or influential, although natural covariates still provide the main explanatory power of the models. Surface water abstraction is a significant covariate in the LR model but of only minor importance in the RF model. Reservoir storage covariates are not significant or are unimportant in both the LR and RF models for this large-sample analysis. Inclusion of WRM terms improves the performance of some models in specific catchments. The LR models of high BFI catchments with relatively high levels of groundwater abstraction show the greatest improvements, and there is some evidence of improvement in LR models of catchments with moderate to high effluent discharges. However, there is no evidence that the inclusion of the WRM covariates improves the performance of LR models for catchments with high surface water abstraction or that they improve the performance of the RF models. These observations are discussed within a conceptual framework for baseflow generation that incorporates WRM practices. A wide range of schemes and measures are used to manage water resources in the UK. These include conjunctive use and low flow alleviation schemes and hands-off flow measures. Systematic information on such schemes is currently unavailable in CAMELS-GB and their specific effects on BFI cannot be constrained by the current study. Given the significance or importance of WRM terms in the models, it is recommended that information on WRM, particularly groundwater abstraction, should be included where possible in future large-sample hydrological data sets and in the analysis and prediction of BFI and other measures of baseflow.

## 1 Introduction

Baseflow, defined as streamflow fed from the deep subsurface and shallow subsurface storage between precipitation and/or
snowmelt events (Tallaksen, 1995; Price, 2011; Zhang et al., 2017; Singh et al., 2019; Gnann et al., 2019), is a hydrological
phenomenon that represents a whole catchment response to meteorological and other environmental signals (Bloomfield et al., 2011). It is important as it sustains surface flows particularly during relatively dry periods and droughts (Smathkin, 2001; Miller et al., 2016), because it supports ecological flows and ecosystem functioning (Poff et al., 1997; Boulton 2003), and is a factor in regulating streamflow quality and temperature (Jordan et al., 1997; Gomez-Velez et al., 2015; Hare et al., 2021). It integrates the outcomes of a wide range of natural and human-influenced surface and sub-surface catchment processes (Price et al., 2011; Gnann et al., 2019) that include geomorpohological controls related to surface topography (Santhi et al., 2008) and soils processes (Vivoni et al., 2007; Price et al., 2011; Singh et al., 2019; Yao et al., 2021) and (hydro)geological processes that control baseflow (Longobardi and Villani, 2008; Bloomfield et al., 2009; Kuentz et al., 2017; Carlier et al., 2018). Land use and land cover (LULC) change may also have profound effects on baseflow generation (Zhang and Schilling, 2006; Wang et al., 2014), including effects of changing forest cover and agriculture (Juckem et al., 2008; Ahiablame et al., 2017; Zhang et al., 2017) and urbanization (Simmons and Reynolds, 1982; Chang 2007; Dow, 2007; McGrane 2015). Through these processes, the dynamics of baseflow generation is modulated by meteorological variability over a range of spatial and temporal scales (Beck et al., 2013; Van Loon and Laaha, 2015; Longobardi and Van Loon, 2018) including large-scale circulation patterns (Cheng et al., 2021). There is also growing evidence for the potential impact of climate change on baseflow across a variety of climate and catchment settings (Wang et al., 2014; Ficklin et al., 2016; Ahiabalme et al., 2017; Zhang et al., 2019) and it has been proposed that this should be viewed in the context of increasing sensitivity of changes in droughts and low flows to wider anthropogenic influences (Van Loon et al., 2016; Sankarasubramanian et al., 2020).

Despite this extensive work on baseflow generation dynamics, Gnann et al., (2019) observed that there is still no general theory to explain variations in baseflow between catchments despite the strong evidence that it is largely controlled by the interaction of climate and landscape processes. They explored the role of climate in baseflow generation using baseflow data from the United States of America (USA) and the United Kingdom (UK) and found that in humid settings baseflow can be highly variable due to variations in catchment storage and wetting potential, whereas in more arid settings baseflow has much lower variability and is primarily controlled by vaporization limits. In a complementary study of 435 catchments across the contiguous US and the UK, Yao et al., (2021) found that soil water storage capacity is an important control on baseflow and that generally, BFI increases with storage capacity for a given a climate condition and decreases with aridity for a given storage capacity.

In addition to climate and catchment controls on baseflow, there is evidence that baseflow may be impacted by water resource management (WRM) practices. Here WRM practices is a loosely defined term that encompasses a wide range of activities related to the management of groundwater and surface water resources that are specifically distinct from wider 'human influences' or 'human activities' (Zhang et al., 2019; Mo et al., 2021) that affect LULC, such as of urbanization,

deforestation, and land-management practices. Wang and Kai (2009) referred to WRM practice as 'direct human interferences'. Some examples of WRM practices include abstraction and discharge, changes in conveyance of streams due to changes in channel structure for example for damming, flow regulation and flood management, and development of structures for water storage within catchments including dams and artificial wetlands.

Using a baseflow recession analysis, Wittenburg (2003) identified reduced baseflow resulting from abstraction for summer irrigation in a catchment in Turkey but only saw limited effect of abstractions for agricultural irrigation on baseflow in a catchment in Germany. The latter was attributed to the location of the abstractions within the catchment (abstractions were primarily near the watershed) and that the abstracted groundwater was not entirely lost to the groundwater balance (with lowered evapotranspiration stress, relative to the Turkish case study, associated with the irrigation contributing to recharge).

Using an empirical analysis of baseflow recession, Wang and Cai (2009) modelled the impact of abstraction and effluent returns on stream flow in a catchment in Illinois, USA. They found that the WRM practices significantly altered recession process and low-flow hydrograph characteristics (compared with land-use change process that affected both the rising and falling limbs of the hydrograph and peak flows) and showed that effluent returns caused a significant increase in low-flow (Q5) magnitude but a decreased low flow variability. In a statistical analysis of trends in baseflow in a catchment in Florida,

USA, Webber and Perry (2006) documented a long-term decline in baseflow and spring flows. They assessed the possible effects of changes in rainfall, LULC and groundwater abstraction but concluded that the primary cause of decline in baseflow and spring flow was over-abstraction of groundwater. Thomas et al. (2013) emphasised the importance of taking 'human interference' into account when estimating the baseflow recession constant after documenting higher baseflow recession constants associated with groundwater withdrawals from catchments in New Jersey, USA. They also noted that the location,

size and degree of confinement of abstractions effected the degree to which streamflow was impacted. Large abstractions of groundwater close to streams resulted in larger impacts on streamflow than smaller abstractions from more distant locations, and abstractions from unconfined aquifers had larger impacts than from confined aquifers. A number of modelling studies have simulated the impact of abstraction and other WRM practices on baseflow (Kirk and Herbert, 2002; Parkin et al., 2007; Sanz et al., 2011; de Graff et al., 2014). For example, de Graffe et al., (2014) calibrated the PCR-GLOBWB model with a

dynamic allocation scheme to simulate surface water and groundwater abstractions and corresponding feedbacks. They found impacts of WRM were experienced during periods of low flows when the contribution of groundwater through baseflow is the largest and that return flows changed the timing and duration of the low flow periods, causing baseflow to be maintained for longer. In summary, as with natural controls on baseflow (Gnann et al., 2019), there is as yet no general theory to explain the effects of WRM practices on baseflow, and the effect of a given WRM practice on baseflow may be contingent on a range of

factors including climate, (hydro)geological setting, location, and timing of the activity.

To date, there have been no large-sample, data-led analyses of the impacts of WRM practices on baseflow. This is despite new opportunities being offered to investigate and quantify catchment processes through open access, large-sample hydrology datasets (Addor et al., 2020). Such datasets have been used to provide insights into catchment processes and functioning across multiple climate and catchment settings (Beck et al., 2013; Ochoa-Tocachi et al., 2016; Fouad et al., 2018;

Gnann et al. 2019; Dudley et al., 2020). CAMELS-GB, a recently published large-sample hydrology dataset for Great Britain (GB) (Coxon et al., 2020a; 2020b), is unusual in that it contains quantitative information on WRM practices including surface water and groundwater abstractions, discharges, and reservoir numbers and capacities at the catchment scale. The aim of the present study is to use the CAMELS-GB large-sample dataset to identify which, if any, of these WRM activities influence baseflow; to assess the importance of these activities in the context of other factors known to influence baseflow, such as meteorology, catchment hydrogeology, catchment physiography, and LULC (Price, 2011); and, to investigate if WRM factors are important in any particular catchment or management settings. More generally, this study also directly addresses 'Challenge 2' of Wagener et al., (2021) related to the need to improve understanding of the impact of human activities on the water cycle in GB.

As Price (2011) has noted, there are four broad approaches to quantify baseflow, as follows: low flow event time series; flow-duration statistics; baseflow recession analysis; and, metrics of the proportion of baseflow to total flow, also known as baseflow indices. This study takes the last approach and specifically uses the two measures of Baseflow Index (BFI) reported in CAMELS-GB (Coxon et al., 2020a; 2020b). BFI is the ratio of baseflow volume to total flow volume expressed as a fraction (Nathan and McMahon, 1990) and can be estimated by hydrograph separation using a wide range of tracer-based and non-tracer methods (Eckhardt, 2008; Gonzales et al., 2009; Price et al., 2011). The two measures of BFI in CAMELS-GB both use non-tracer-based methods, specifically a digital filter (Lyne and Hollick, 1979) and a graphical / statistical method (Gustard et al., 1992; Piggot et al., 2005). The former, although it is not based on the physics of discharge processes, produces objective and reproducible estimates of BFI (Cheng et al., 2021), while the latter has been used previously to characterise BFI across the study area (Bloomfield, 2009).

Two statistical models (multiple linear regression, LR, and machine learning using random forests, RF) are used here to investigate the relationships between the two estimates of BFI and WRM and other catchment covariates in the CAMELS-GB dataset. Although studies of BFI typically consider multiple baseflow filters to reduce uncertainty in estimates of BFI (Chen and Teegavarapu, 2020; Kissel and Schmalz, 2020; Zhang et al., 2020), the present study is not designed either to assess the relative efficacy of the filters used to estimate BFI, nor to compare the respective efficacy of the chosen statistical models in estimating baseflow: this is not a model inter-comparison study (Refsgaard and Knudsen, 1996). Instead, the estimates of BFI and the modelling approaches are designed to provide complementary evidence for the nature and importance (or not) of WRM practices on influencing BFI based on the published CAMELS-GB data.

## 2 Study area and data

### 2.1 Study area

This study focuses on 429 catchments across GB (Fig. 1) covering a wide range of climate-landscape-water management features (Fig. 2). Catchments in north and north-west of the study area tend to have higher mean elevations than those in the south and south-east (Coxon et al., 2020a). Meteorology tends to reflect the broad gradient in catchment physiography, with

wet and cooler conditions typically prevalent in the north and west of the study area compared with relatively dry and warmer conditions in the south-east (Fig. 2a). The dominant land cover also reflects the prevailing physiographic and meteorological conditions with grass cover predominating in the north and west and crop cover in the south and east, with urban land-cover dominant in London and the other large cities of central and northern England (Fig. 2b). High productivity aquifers are found in the south-east and east (Fig. 2c; Bloomfield et al., 2009; Marchant and Bloomfield, 2018), whereas less productive aquifers and non-aquifers are generally more extensive in the west and north-west. Catchments in which clay dominated soils overlie mudrock and clay bedrock formations and catchments with extensive glacial till deposits that are present in central and eastern areas (Fig. 2d) (Bloomfield et al., 2009; Bricker and Bloomfield, 2014).

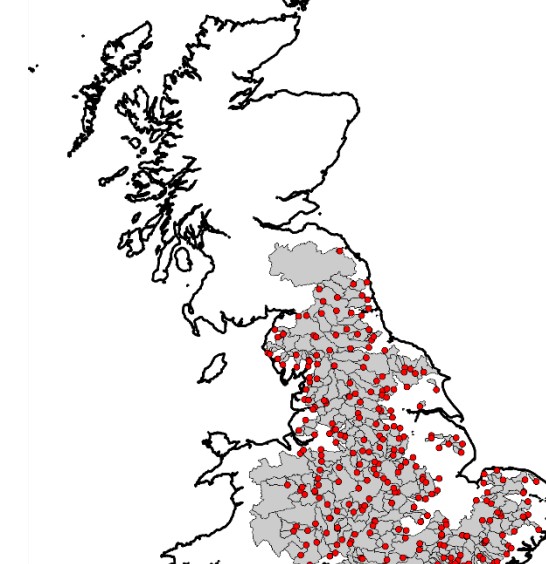

[contains OS data © Crown copyright and database right 2021]

**Figure 1.** Location of catchments in the study area

Groundwater is used throughout England and forms on average about 30% of the public supply, as well as being used extensively for agricultural irrigation and industrial supplies (Ascott et al., 2017). For 2017 (the last year of published abstraction data) abstractions from all sources (except tidal) in England totalled 10,395 million cubic metres (Mm$^3$), with 8,350 Mm$^3$ from surface waters and 2,044 Mm$^3$ from groundwater. Just over half of all these abstractions were used for public supply (5,332 Mm$^3$) (UK Government, 2020). Regionally groundwater use is more important in southern and eastern England where groundwater abstraction may consist of 100% of public supply (Ascott et al., 2020). Consequently, there is a tendency for

more extensive surface water abstraction in the north and more groundwater abstraction in south-east (Fig. 2e and 2f) (Coxon et al., 2020b). Effluent discharges are generally relatively high in catchments in and near major urban centres such as London, central England, and across parts of the north-west (Fig. 2b and 2g) while the highest reservoir capacity is generally associated with catchments in northern and western parts of the study region (Fig. 2h).

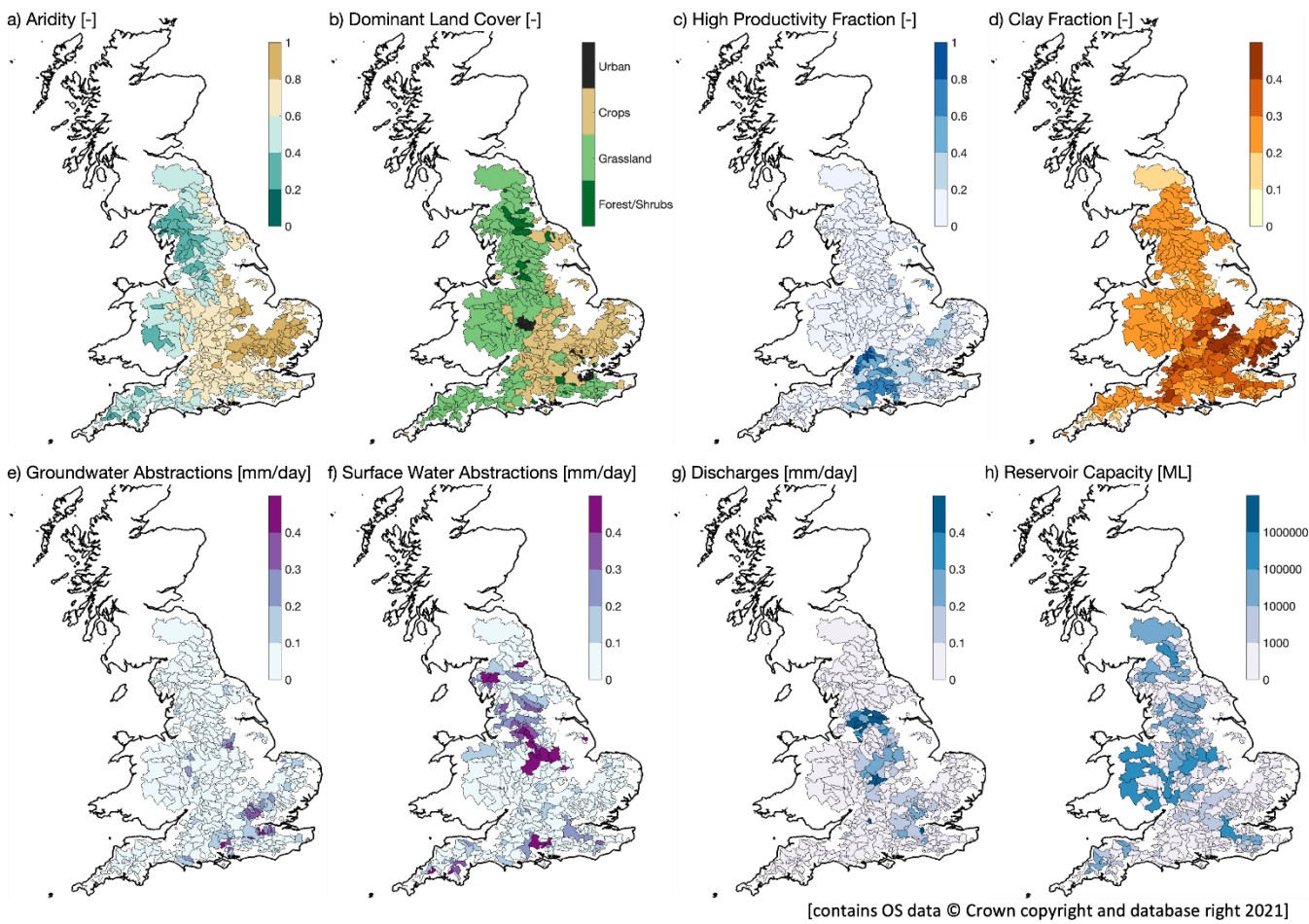

[contains OS data © Crown copyright and database right 2021]

**Figure 2.** Selected catchment characteristics from CAMELS-GB.

**2.2 Data**

The data used in this study have been taken from the CAMELS-GB large-sample hydrology data set for Great Britain (GB) (Coxon et al., 2020a, 2020b), itself part of the wider CAMELS (Catchment Attributes and MEteorology for Large-sample 160     Studies) initiative (Newman et al., 2015; Addor et al., 2017; 2020; Alvarez-Garreton et al., 2018; Chagas et al., 2020). CAMELS-GB is unique in that it contains human-influence attributes for some catchments, and it is that sub-set of catchments

which are used here. These initially consisted of 442 catchments for which there are 'human influence attributes' (Coxon et al., 2020a, Table2). However, these were further reduced to 429 catchments (Fig. 3) following a consideration of the estimates of BFI that are available for those catchments and the availability of data for the covariates of interest, as described below.

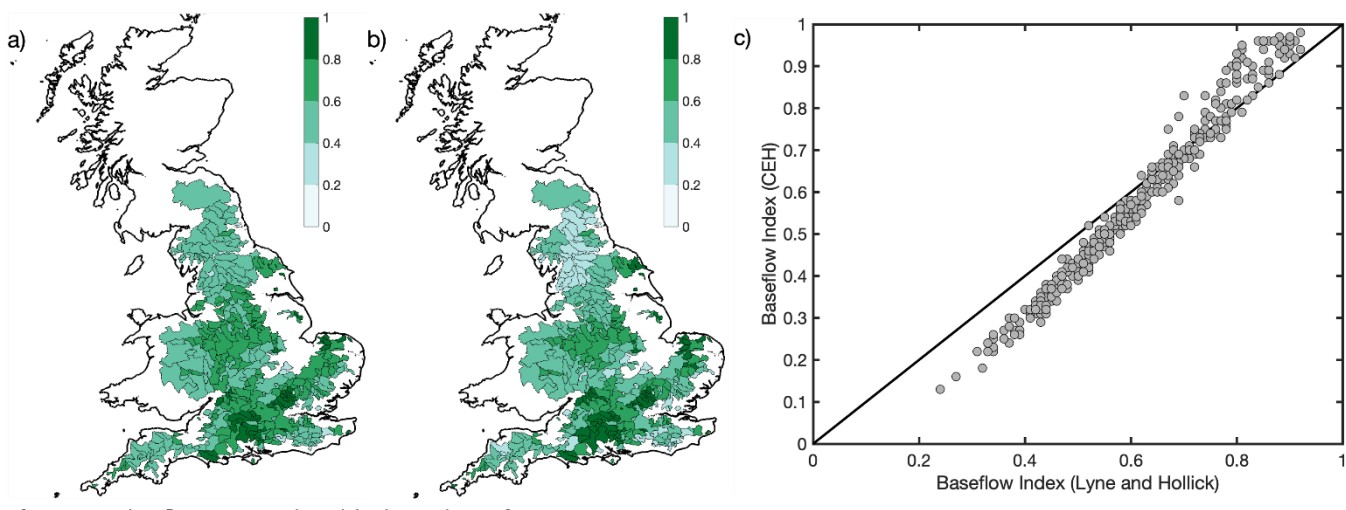

[contains OS data © Crown copyright and database right 2021]

**Figure 3.** The distribution of a. BFI_CEH, b. BFI_LH, and c. relationship between the two measures of BFI with a 1:1 line for reference.

BFI is a hydrological signature (Price et al., 2011; McMillan, 2021) that can be estimated using a wide range of techniques. CAMELS-GB contains two estimates of baseflow. One index, 'baseflow_index_ceh' (BFI_CEH) (Fig3a.), is derived using a method developed by the UK Centre for Ecology & Hydrology and has been used in previous studies of baseflow and flow regimes in Great Britain (Gustard et al., 1992; World Meteorological Organization, 2008). The other, 'baseflow_index' (BFI_LH) (Fig. 3b), was estimated by baseflow separation using the Lyne and Hollick digital filter (Lyne
and Hollick, 1979) as implemented by Ladson et al., (2013). A comparison of the two CAMELS-GB baseflow indices (Fig. 3c) confirms the common observation that different techniques used for baseflow separation influence the estimated indices (Nathan and McMahon, 1990; Eckhardt, 2008; Beck et al., 2013, Addor et al., 2017). There are often large uncertainties in the underlying streamflow data used to estimate BFI (Coxon et al, 2015) but these are difficult to characterise across large samples of catchments and uncertainty estimates are not available for all the CAMELS-GB catchments (Coxon et al., 2020b). However,
BFI typically has lower uncertainty compared with other hydrological signatures, as it is based on temporal averaging (Westerberg and McMillan, 2015), and that only typically small differences in the BFI estimates are observed in the present study based on the two methods of estimate (Fig. 3).

Given that the true BFI for any given catchment is unknown, catchments for analysis in this study have been selected where there is a reasonable agreement between the two baseflow indices. Ten catchments were removed where there is an

absolute difference between BFI_CEH and BFI _LH of greater than 0.14, equivalent to the largest 2.5%tile of the absolute differences of the population. A further three catchments were removed due to missing covariate data leaving the 429 catchments for analysis (Figs. 1 and 3). Coxon et al. (2020b) note that the CAMELS-GB baseflow indices have been estimated for flow time series available during water years from $1^{st}$ Oct 1970 to $30^{th}$ Sept 2015, but that individual time series lengths and completeness may vary between catchments. On average, flow records for the 429 catchments are 91% complete with only 48 catchments with <75% complete. No sites have been omitted from the analysis based on the length of their flow records. Figure 3c shows that there is a generally good linear agreement between the two estimated BFI indices. However, for BFIs below 0.7 BFI_CEH is systematically lower than BFI_LH, and for BFIs above 0.7 BFI_CEH is systematically higher than BFI_LH. In addition, for sites above a BFI of about 0.7 the correlation between the two indices is reduced.

21 of the CAMELS-GB catchment attributes (Coxon et al., 2020a) related to catchment physiography, climate, hydrogeology, land cover and soils as well as WRM practices have been selected as covariates for analysis (Table A1). The spatial distribution of selected covariates are provided in Fig. 2 and described in Coxon et al. (2020b). The 21 CAMELS-GB covariates used in this study have been selected to be representative of each of the major components in a new conceptual model of baseflow generation (Fig. 4) and are consistent with the recently proposed, broader perceptual hydrological model for GB (Wagener et al., 2021). Five WRM covariates from the CAMELS-GB dataset have been selected for analysis: groundwater abstraction (groundwater_abs), surface water abstraction (surfacewater_abs), effluent discharges (discharges) to streams and the number and capacity of reservoirs within catchments (num_reservoirs and reservoir_cap). Note that the discharge term only accounts for effluent from sewage treatment works and does not provide information on other water returns (Coxon et al., 2020b). Price (2011) presented a conceptual model that illustrated how components of the terrestrial water cycle and specific catchment processes are related to baseflow based on stores and flows of water in catchments. It did not, however, incorporate WRM concepts and how these might influence or modify baseflow. In addition, it did not include aspects of catchment physiography as it focussed on catchment inputs, storage and losses. Fig. 4 is a revised conceptual diagram (building on Price et al, 2011) indicating conceptual relationships between baseflow, catchment compartments and processes that lead to baseflow generation, including aspects of WRM. It conceptualises WRM practices as simple high-level flows between groundwater, streamflow and components of storage. Some flows that may be significant within a given catchment, such as mains leakage (conceptualised Fig. 4), however these are outside the current analysis as there is no information for these flows in CAMELS-GB.

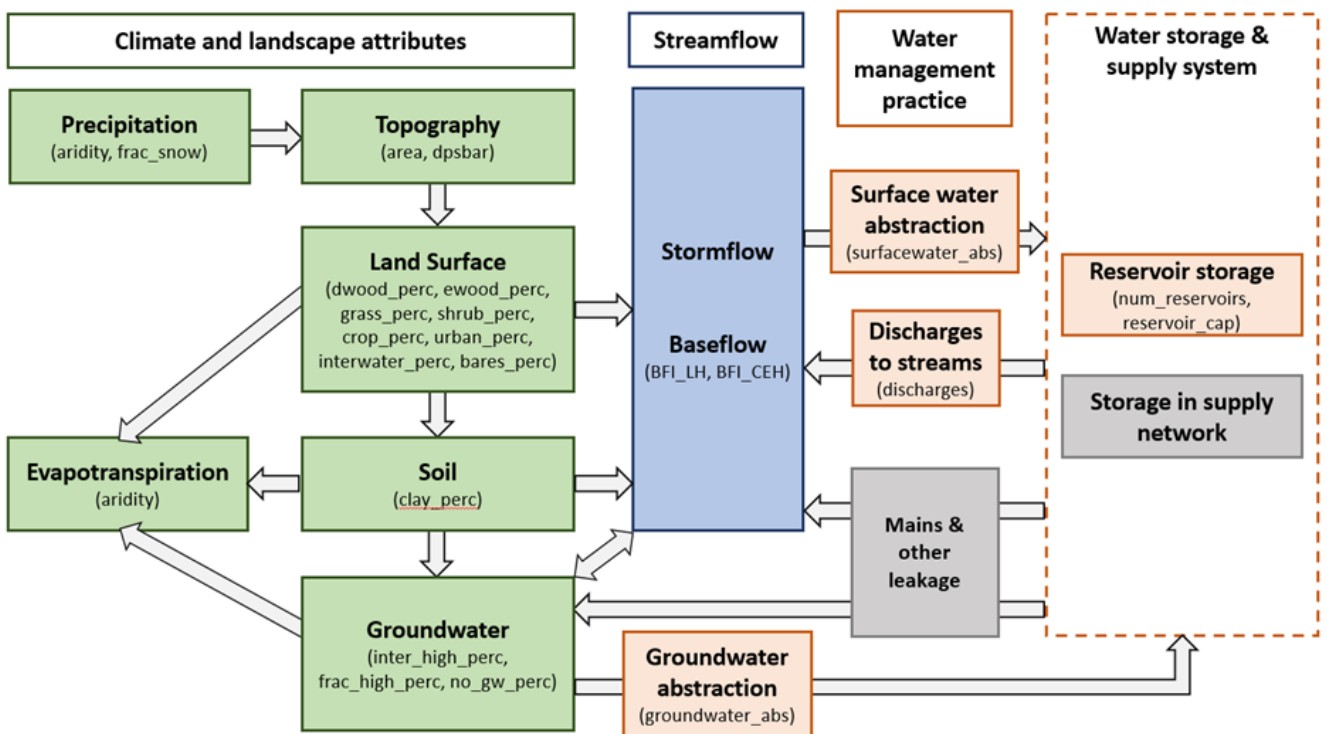

**Figure 4.** Conceptual model of the relationships between the major compartments of the terrestrial water cycle that exert an influence on baseflow. Baseflow and storm flow components are highlighted in blue, driving climatology, catchment characteristics and compartments are shaded in green, and human influences within the conceptual model are shaded in orange and grey (the latter outside the scope of this study). The 21 CAMELS-GB covariates and the two BFI parameters used in this study are listed against their respective compartments within the conceptual framework.

## 3. Modelling methods

Modelling is used in this study not for predictive purposes but to explore model structures and performance to assess the evidence for the relative importance (or not) of WRM practices in influencing BFI. Two complementary modelling schemes, a multiple linear regression (LR) scheme and a random forest scheme (RF), have been applied to two estimates of BFI (BFI_LH and BFI_CEH) using two sets of covariates (Set A and Set B). Set A consists of the 16 natural covariates and Set B consists of all 21 CAMELS-GB covariates, i.e. the combined natural and human influence covariates (Table A1). Consequently, eight models (Models 1 to 8) have been developed and evaluated. The LR and RF models are first calibrated for the Set A covariates (Models 1 to 4), then a second separate calibration is undertaken using Set B covariates (Models 5 to 8). The resulting model structures are investigated and their performance in estimating observed BFI compared without and with WRM covariates to understand the influence of WRM covariates on BFI.

The accuracy of the model estimates has been assessed using RMSE and by calculating Lin's concordance coefficient (Lin, 1989) for the predicted and measured values. Lin's coefficient indicates the degree of similarity between two variables, where

$$\rho_c(x,y) = \frac{2\rho(x,y)\sqrt{\text{var}(x)}\sqrt{\text{var}(y)}}{\text{var}(x)+\text{var}(y)+(\mu_x-\mu_y)^2},\tag{1}$$

and where $\rho_c(x,y)$ is Lin's concordance coefficient for variables $x$ and $y$, $\rho(x,y)$ is Pearson's coefficient for the same variables, $\text{var}(x)$ is the variance of $x$, and $\mu_x$ in the mean of $x$. Lin's concordance coefficient can take values between -1 and 1. A value of 1 indicates an exact match between the two variables and the $(\mu_x - \mu_y)^2$ term means that variables with different mean values have a small coefficient value in contrast to standard correlation coefficients where perfectly correlated variables can have vastly different mean values. Lin's concordance coefficient is in contrast to a more standard Pearson correlation coefficient that is an indication of the explanatory power of a linear relationship between the two variables. Lin's concordance coefficient is calculated both to assess the accuracy of a given model at replicating the training data and in a 10-fold cross-validation procedure to explore the model accuracy at locations that were not used in calibration. If Lin's coefficient is substantially smaller upon cross-validation then this could be an indication that the model is overfitted.

### 3.1 Linear regression

Regression is commonly used to model the effect of a given set of covariates on a variable of primary interest (Fahrmier et al., 2013). Here generalised linear regression (Dobson, 2002) is used to investigate the relationship between BFI_LH and BFI_CEH and the 21 catchment covariates. Logit transformation was applied to the BFI data, as $y_i = \log(z_i/(1 - z_i))$, where $z_i$ is the BFI of catchment $i$. This is to ensure the fitted, back transformed BFI values are constrained between 0 and 1.

The model considered in this paper is a linear mixed model with the following form,

$$Y = X_1\beta_1 + \cdots + X_p\beta_p + \epsilon \qquad \epsilon \sim \mathcal{N}(0, \sigma^2 R)\tag{2}$$

where $Y = (y_1, \ldots y_n)'$ denotes the column vector of BFI values from $n$ catchments, $X_j = (x_{j1}, \ldots, x_{jn})', j = 1, \ldots p$, are the column vectors of the covariates (catchments attributes). The column vector $\epsilon$ represents the model residuals, which are assumed to follow a normal distribution, with covariance matrix $\sigma^2 R$, where $R$ reflects the correlation between transformed BFI values. The linear sums of covariates in a linear mixed model are referred to as the fixed effects and the residual term as the random effects.

In this paper, the model parameters $\beta = (\beta_1, \ldots, \beta_p)'$ are estimated using the generalized least squares estimator (Dobson, 2002):

$$\beta = (X'R^{-1}X)^{-1}X'R^{-1}Y\tag{3}$$

These parameter values maximise the likelihood or probability that the data would have arisen from the estimated model. Standard linear regression requires the assumption that the residuals are independent and identically distributed (iid) and that

the correlation matrix is equal to the identity matrix, $I$. Such an assumption can be inappropriate for landscape measurements, as they are not selected according to a randomised design, and are often correlated in space as a result of the underlying geology, climate, etc. In particular, the BFI measurements made from locations closer to each other are more likely to share some similarity than those a long distance apart. If this correlation is ignored then the significance of some model terms could be exaggerated.

A further issue is deciding which of the available covariates should be included. If too few covariates are included then some of the key drivers of BFI variation might be missed and the predictions that result might be imprecise. If too many covariates are included then the model might be overfitted. Some of the terms in an overfitted model replicate the random variation of the BFI values within the calibration data rather than generally applicable relationships between BFI and the covariates. Such a model can accurately predict the BFI for the sites used in calibration but performs less well on other data.

The addition of a covariate to a model generally increases the maximised likelihood even in the absence of a true relationship between that covariate and the property of interest. The addition cannot decrease this likelihood because the original model can be achieved if $\beta_{p+1} = 0$. A statistical criterion must be used to decide whether the increase in likelihood upon the addition of a parameter is sufficient to justify the inclusion of that term.

        The modelling procedure consists of three steps. In the first step, given the candidate covariates, variable selection is

carried out using the stepwise selection routine based on the Akaike Information Criterion (AIC; Akaike, 1973). The AIC, which is twice the negative log-likelihood of the model minus 2 times the number of model parameters,

$$AIC = -2 \log\{\mathcal{L}(\beta, \sigma^2; Y)\} - 2(p + 1) \qquad (4)$$

The model with the lowest AIC is considered to be the best compromise between accuracy and complexity. The forwards selection routine starts with a model containing no covariates. Each candidate covariate is considered in turn and the AIC that results from its addition to the model is recorded. The covariate which leads to the largest decrease in AIC is added to the

model. The iterative procedure continues until none of the remaining covariates lead to a decrease in AIC. This procedure is initially conducted assuming independent residuals (i.e., $R = I$) and is implemented using the "step" function from R package "stats". In the second step, spatial correlation is assessed by calculating empirical variograms (Cressie, 1993) of the model residuals using the "variogram" function from R package "gstat". The variogram indicates how the expected squared difference

between a pair of residuals varies according to their distance apart. Finally, a model including spatial correlation in the residuals is estimated when inspection of the variogram indicates that this is necessary. Specifically, the spatial correlation is reflected by the non-zero off-diagonal elements in the correlation matrix, $R$ which correspond to the values from an exponential correlation function (i.e., $r(d_{ij}) = \exp(-d_{ij}/\varphi)$, where $d_{ij}$ is the Euclidean distance between two catchments $i$ and $j$ and $\varphi$ is an estimated model parameter). The model with spatial correlation can be estimated by residual maximum likelihood

(REML; Lark et al., 2006) using the "gls" function from R package "nlme". The statistical significance of each covariate included in the model (i.e. whether the corresponding regression coefficient is significantly different to zero) is recorded for p values of 0.1, 0.05 and 0.001.

### 3.2 Machine learning

LR models require assumptions about the nature of baseflow variation that can restrict the patterns of variation which the
model can represent. In the past few decades, machine learning (ML) methodologies have become increasingly popular for representing complex environmental variation (e.g. Hengl et al., 2018; Lange and Sippel, 2020; Nearing et al 2020). ML algorithms lead to considerably more flexible relationships between environmental variables. For example, regression trees recursively partition observation locations according to a series of binary tests on their covariate values. Each location enters the tree at the initial decision node and then follows one of two branches according to the result of the initial test. Each branch
leads to a network of further decision nodes and tests until the location is allocated to a terminal node. The predicted value of the environmental variable at an unobserved location is equal to the average of the training data that are allocated to the same terminal node. The tests at each node are optimised so that the total squared errors for a tree of a specified degree of complexity is minimised.

Regression trees can replicate complex nonlinear relationships that include interactions between different covariates
but they are prone to overfitting. A regression tree can perfectly predict the variable of interest for some training data if the number of terminal nodes is equal to the number of training observations but it cannot be expected to perform exactly when predictions are made at other locations. Overfitting can be reduced by introducing stopping criteria to the trees (e.g. each terminal node must contain a specified proportion of the training data) or by using cross-validation to decide whether a particular decision node should be included in the tree. Overfitting might be further reduced by combining an ensemble of
regression trees to form a random forest (Breiman, 2001). The trees within the ensemble differ because they are estimated for a different bootstrap sample of the available data and a different subset of the candidate covariates is considered at each decision node. The prediction of the variable of interest at a particular location is equal to the average prediction across all the trees. Addor et al. (2018) found that the inclusion of 500 trees in a random forest considerably stabilised predictions and smoothed relationships between their covariates and BFI measurements.

The random forest interprets the available data as if they were a random sample of the population of interest and does not account for spatial correlation amongst the observations. Also, the relationships implied by a random forest model cannot be stated in a simple parametric form such as Eq. (1) meaning that it can be a challenge to determine the drivers of variation. It is possible to assess the importance of each covariate by shuffling the values of that covariate amongst the observation locations and calculating the reduction in prediction accuracy. However, Schmidt et al. (2020) and Wadoux et al. (2020) advise
caution when inferring causal relationships from random forest models. Wadoux et al. (2020) demonstrate that photographs of soil scientists projected across their study area can be utilised by a random forest to accurately map the soil carbon content. They suggest that knowledge discovery from ML models requires more than the recognition of patterns and successful prediction. Instead they recommend the pre-selection of relevant environmental covariates and the posterior interpretation and evaluation of the recognised patterns: this is the approach taken here with the selection of 21 covariates representative of the
conceptual framework being analysed (Fig. 4).

Random forests are calibrated using the Matlab 'Treebagger' function with each forest containing 500 trees (consistent with Addor et al., 2018), the with-replacement bootstrap sample for each tree being of the same size as the set of available data and one third of the covariates are considered at each decision node. The 'Treebagger' function defines the importance of a covariate in a random forest to be equal to the increase in the mean squared error of all predictions averaged over all trees in the ensemble upon shuffling of the covariate values divided by the standard deviation of the predictions taken over the trees.

## 4. Results

### 4.1 Linear regression model structures

Regression models were developed for both BFI_LH and BFI_CEH, with covariates from Set A (Models 1 and 2) and from Set B (Models 5 and 6). For all four models the variograms of the residuals indicated substantial spatial correlation. Therefore, the models were re-estimated by REML and included spatial correlation parameterised by an exponential function. Note that although the inclusion of the residual correlation structure does not alter signs of the estimated coefficients, the significance of the model covariates changed. Some covariates were no longer significant after accounting for the spatial correlations. This could imply that part of the variation in BFI that was previously explained by certain covariates in the iid model may have been a result of spatial correlation. The full LR models are listed in Table A2 and the distribution of residuals for the LR models are illustrated in Fig. A1.

Figure 5 shows the covariates identified as significant as well as the sign of the covariates. In this analysis, topography ("dpsbar"), climate ("aridity"), the spatial coverage of fractured aquifers ("frac_high_perc"), of crop coverage ("crop_perc") and of clay soils ("clay_perc") are highly significant in all four LR models, and the spatial coverage of areas with no active groundwater systems ("no_gw_perc") is also a significant covariate in all four models to different levels of significance (Fig. 5). In the LR models using Set B (Models 5 and 6), surface and groundwater abstractions and effluent discharges are all highly significant in explaining the variations in the BFI_LH and BFI_CEH although the number ("num_reservoirs") and capacity of reservoirs ("reservoir_cap") are not significant covariates. Urban land cover ("urban_perc"), previously noted as potentially influencing BFI in the Thames Basin in southern England (Bloomfield et al., 2009), is not a significant covariate in the LR models using covariate Set A once spatial correlation in the covariates has been accounted for, and is not significant at all when WRM covariates are include in the LR models.

In the LR models, the signs of the significant natural covariates in Fig. 5 (Models 1 and 2) are consistent with current process-based understanding of the generation of baseflow (Price et al., 2011; Gnann et al., 2019; Yao et al., 2021) as represented in the revised conceptual model (Fig. 4) and with previous regression models of BFI in the study area (Bloomfield et al., 2009). For example, there is a significant inverse relationship between BFI and the fraction of clay soils within catchments, the fraction of catchments underlain by rocks with essentially no groundwater, and the aridity of catchments.

Conversely, all LR models indicate a significant positive correlation between BFI and the fraction of catchments underlain by fractured aquifers.

In all four LR models, the Lin's concordance coefficients between the fixed effects predictions and the observed BFI are similar upon training and validation indicating that the models are not overfitted (Table 1). The coefficients for the models using Set A (Models 1 and 2) to predict BFI_LH and BFI_CEH are 0.75 and 0.81 respectively. There are moderate negative correlations between the residuals from these models and the surface water and groundwater abstractions and effluent discharges from Set B covariates (Table 2). There are negligible correlations between the residuals and the number and capacity of reservoirs covariates. When the WRM covariates are added to the model (Models 5 and 6) the Lin's concordance coefficients increase to 0.82 and 0.85 for BFI_LH and BFI_CEH respectively (Table 1).

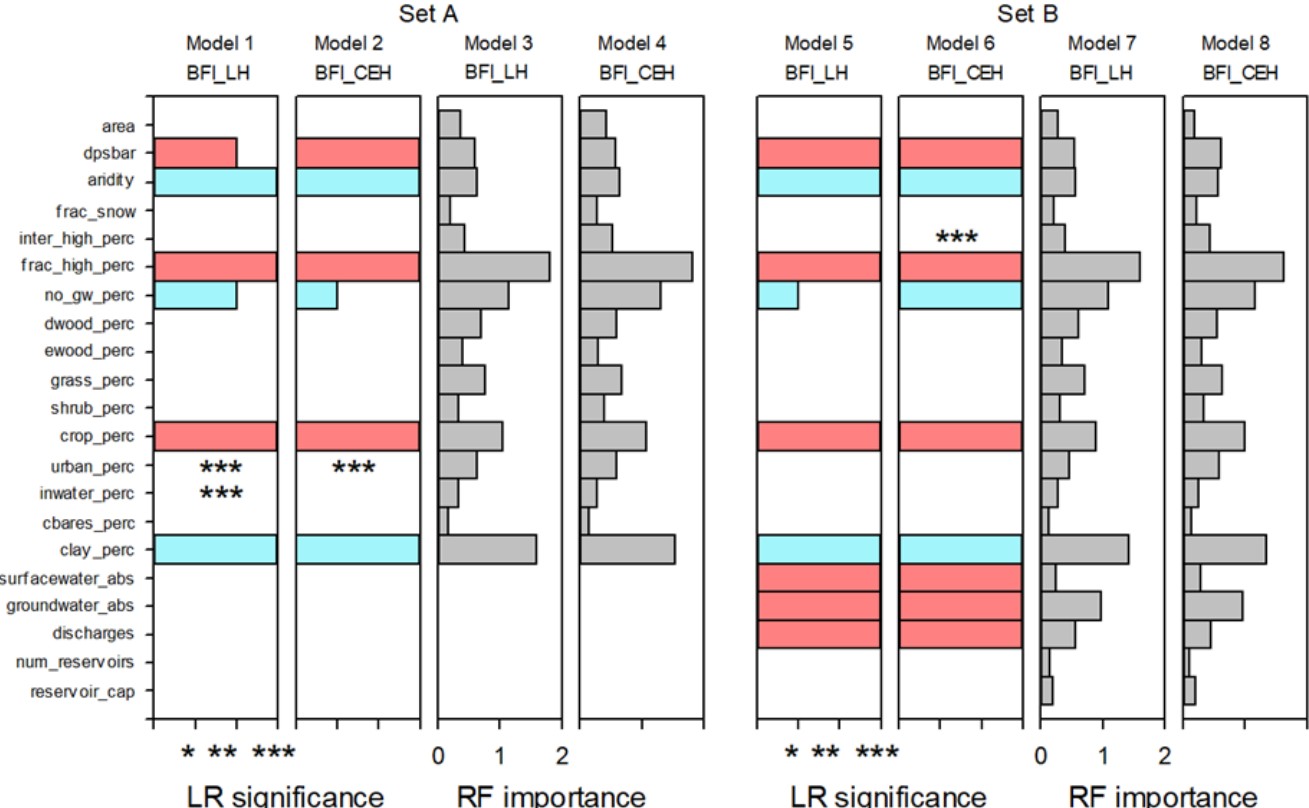

**Figure 5.** Signs and significance levels of the covariates in the LR models and the relative importance of covariates in the RF models. The signs of the significant covariates in the LR models are indicated using colour (pink for positive, blue for negative) and the corresponding significance levels of the covariates are indicated on the x-axis with asterisks (* for significance level between 0.05 and 0.1, ** for significance level between 0.01 and 0.05, *** for significance level below 0.001). Some covariates were only significant prior to accounting for the spatial correlations. These are marked with asterisks only in the figure at their respective level of significance. Table A1 gives full details of the regression coefficients. Relative RF importance ranges from zero to 2. Table 3 gives the scores for the relative importance of covariates in the four RF models.

**Table 1.** Lin's concordance coefficients between LR model predictions and the data.

| Model | Model 1 | Model 2 | Model 5 | Model 6 | Model 3 | Model 4 | Model 7 | Model 8 |
|---|---|---|---|---|---|---|---|---|
| Scheme | LR | LR | LR | LR | RF | RF | RF | RF |
| | Set A | Set A | Set B | Set B | Set A | Set A | Set B | Set B |
| BFI data | BFI_LH | BFI_CEH | BFI_LH | BFI_CEH | BFI_LH | BFI_CEH | BFI_LH | BFI_CEH |
| Training | 0.75 | 0.81 | 0.82 | 0.85 | 0.95 | 0.96 | 0.97 | 0.97 |
| Validation | 0.75 | 0.80 | 0.82 | 0.84 | 0.80 | 0.82 | 0.81 | 0.84 |

**Table 2.** Pearson correlation between Set A model residuals and Set B model covariates

| Model Scheme | LR | LR | RF | RF |
|---|---|---|---|---|
| BFI data | BFI_LH | BFI_CEH | BFI_LH | BFI_CEH |
| surfacewater_abs | -0.16 | -0.17 | -0.21 | -0.19 |
| groundwater_abs | -0.36 | -0.31 | -0.27 | -0.27 |
| discharges | -0.27 | -0.23 | -0.16 | -0.13 |
| num_reservoirs | -0.02 | -0.02 | -0.03 | -0.02 |
| reservoir_cap | -0.01 | -0.02 | -0.04 | -0.03 |

In summary, when spatial correlation effects are taken into account, the LR models do not appear to be overfitted, show a consistent though moderate improvement in explanatory power with the addition of the WRM covariates, and indicate that groundwater and surface water abstraction, and effluent discharges are all significant in explaining the variations in both the estimates of BFI.

**4.2 Machine learning model structures**

The relative importance of the covariates with respect to estimates of BFI are listed in Table 3 and illustrated in Fig. 5 for the RF Set A models (Models 3 and 4) and Set B models (Models 7 and 8). Lin's concordance coefficients on training data are larger for the RF predictions than for the LR models (Table 1). However, upon cross-validation the RF coefficients decrease and are comparable to the LR model values. This could be an indication of overfitted RFs, perhaps because the spatial correlation previously identified amongst the data (see LR results above) has not been accounted for in the RF models. The

most important covariates in the RF models using Set A covariates (Models 3 and 4) are consistent for both BFI_LH and BFI_CEH and are in descending order of importance: the fraction of catchments underlain by fractured aquifers ("frac_high_perc"), clay soils ("clay_perc"), extent of catchments underlain by rocks with essentially no groundwater ("no_gw_perc"), and crop and grass coverage ("crop_perc", "grass_perc") (Table 3 and Fig. 5).

The residuals from the RF models are moderately and negatively correlated for the surface water and groundwater
abstraction covariates (Table 2). The groundwater abstraction covariate has high importance in both RF models of Set B
covariates (Models 7 and 8, Table 3 and Fig. 5). The discharges covariate has a moderate importance in the RF models, but
the relative importance of the surface water abstraction covariate and the covariates for the number of reservoirs and for their
total capacity is low (Table 3 and Fig. 5).

**Table 3.** Score of the relative importance of covariates in RF model

| Covariate | Model 3 | Model 4 | Model 7 | Model 8 |
|---|---|---|---|---|
| area | 0.36 | 0.43 | 0.28 | 0.18 |
| dpsbar | 0.59 | 0.58 | 0.54 | 0.62 |
| aridity | 0.63 | 0.64 | 0.55 | 0.57 |
| frac_snow | 0.20 | 0.28 | 0.21 | 0.22 |
| inter_high_perc | 0.43 | 0.52 | 0.39 | 0.44 |
| frac_high_perc | 1.81 | 1.82 | 1.59 | 1.62 |
| no_gw_perc | 1.14 | 1.3 | 1.09 | 1.16 |
| dwood_perc | 0.69 | 0.59 | 0.6 | 0.55 |
| ewood_perc | 0.39 | 0.30 | 0.34 | 0.31 |
| grass_perc | 0.76 | 0.67 | 0.70 | 0.64 |
| shrub_perc | 0.33 | 0.39 | 0.30 | 0.34 |
| crop_perc | 1.05 | 1.07 | 0.88 | 1.00 |
| urban_perc | 0.63 | 0.59 | 0.46 | 0.58 |
| inwater_perc | 0.34 | 0.27 | 0.27 | 0.26 |
| cbares_perc | 0.17 | 0.14 | 0.12 | 0.14 |
| clay_perc | 1.59 | 1.53 | 1.41 | 1.34 |
| surfacewater_abs | 0 | 0 | 0.24 | 0.28 |
| groundwater_abs | 0 | 0 | 0.96 | 0.96 |
| discharges | 0 | 0 | 0.55 | 0.45 |
| num_reservoirs | 0 | 0 | 0.15 | 0.10 |
| reservoir_cap | 0 | 0 | 0.19 | 0.21 |

In summary, RF models show that the majority of the power to explain variations in BFI is due to the natural covariates
and when WRM covariates are included in the models, groundwater abstraction is the most important and effluent discharges
of moderate importance in explaining both estimates of BFI.

## 4.3 Consistency between model structures

The results of the models are subject to standard caveats for such types of analysis. Inclusion of spatial correlation in the LR models was necessary and led to some otherwise significant covariates being removed, and the LR models were unable to represent non-linear relationships between the covariates. The RF models did not take into account spatial correlation identified in the LR analysis and there was some evidence of overfitting of the RF models, but they are able to represent any non-linearities that are present between the covariates that could not be included in the LR models. Notwithstanding these observations, the two contrasting modelling approaches, one relatively simple and tractable (LR modelling) and the other considerably more flexible but potentially harder to interpret (RF modelling), have resulted in remarkably similar model structures with high levels of consistency between both natural and WRM covariates being identified as either significant (LR models) or important (RF models).

The structures of the LR and RF models (Fig. 5) are broadly insensitive to the BFI being modelled. Although this is reasonable given the correlation between BFI_LH and BFI_CEH (Fig. 3), this observation supports the inference that the models are robust. Importantly for the purposes of the present study, significant covariates in the LR models and covariates with relatively large importance in the RF models are consistent regardless of whether the models are developed using BFI_LH or BFI_CEH (Fig. 5).

There is a high level of agreement between the two modelling approaches regarding the significance or importance of the natural covariates in Set A. Both the LR and RF models indicate the primary importance of the presence of fractured aquifers in controlling BFI. This is consistent with the observation of Bloomfield et al (2009) where the percentage coverage of fractured aquifers in the Thames catchment in southern GB was found to be an important term in LR models of BFI. In the present study, in Models 1 to 4 the catchment fraction underlain by fractured aquifers is either a significant covariate or the covariate with the largest importance (Fig. 5), and catchment fraction of clay soils, those underlain by rocks with essentially no groundwater, and crop coverage are all significant in the LR models or have large importance in the RF models (Fig. 5). The two other catchment covariates identified as significant in the LR models (topography and aridity) also have moderate importance in the RF models.

The same natural covariates that are identified as significant or of high importance in the LR and RF models in Set A (Models 1 to 4) are also significant or important in models using the Set B covariates (Models 5 to 8) (Fig. 5) and the majority of the variation in BFI is described by the natural covariates (Table A2). From these observations, it is taken that WRM practices, rather than being the principle explanatory factor of variance in BFI, act to modify BFI controlled primarily by natural catchment processes. There are also similarities in the significant or importance of WRM covariates between the Set B models (Models 5 to 8). In both cases groundwater abstraction is significant or important, effluent discharges are significant or of moderate importance, and both reservoir numbers and capacities are either not significant or are of low importance. There is however a notable dissimilarity between the model structures with regard to surface water abstraction: it is a significant

covariate in the LR models (Fig. 5, see Models 5 and 6) but is not important in the RF model (Table 3 and Fig. 5, see Models 7 and 8).

## 4.4 Evidence for the impact of water resources management practices

The observations relating to the effect of WRM on BFI have been investigated further by considering the extent to which particular catchment context and management settings influence the respective model performance. Figure 6 shows that, particularly for a number of relatively high BFI catchments in central England and SE England to the north of London (Fig. 6a), the LR model of BFI_LH using only natural covariates appears to underestimate BFI. Similar observations can be made with respect to estimates of BFI_CEH (Fig. A2a), with the additional observation that there are a few catchments in eastern

England where the model appears to overestimate BFI. Inclusion of WRM covariates leads to some improvements in LR model estimates of BFI, with the largest improvements being in the high BFI catchments (Fig. 7a and A3a). These improvements are particularly seen in the relatively high BFI catchments immediately to the north of London (Fig. 6b). Note, however, that addition of WRM covariates to the models does not appear to improve the estimates of BFI_CEH in the catchments in eastern England, where the model still appears to overestimate BFI (Fig. A2b).


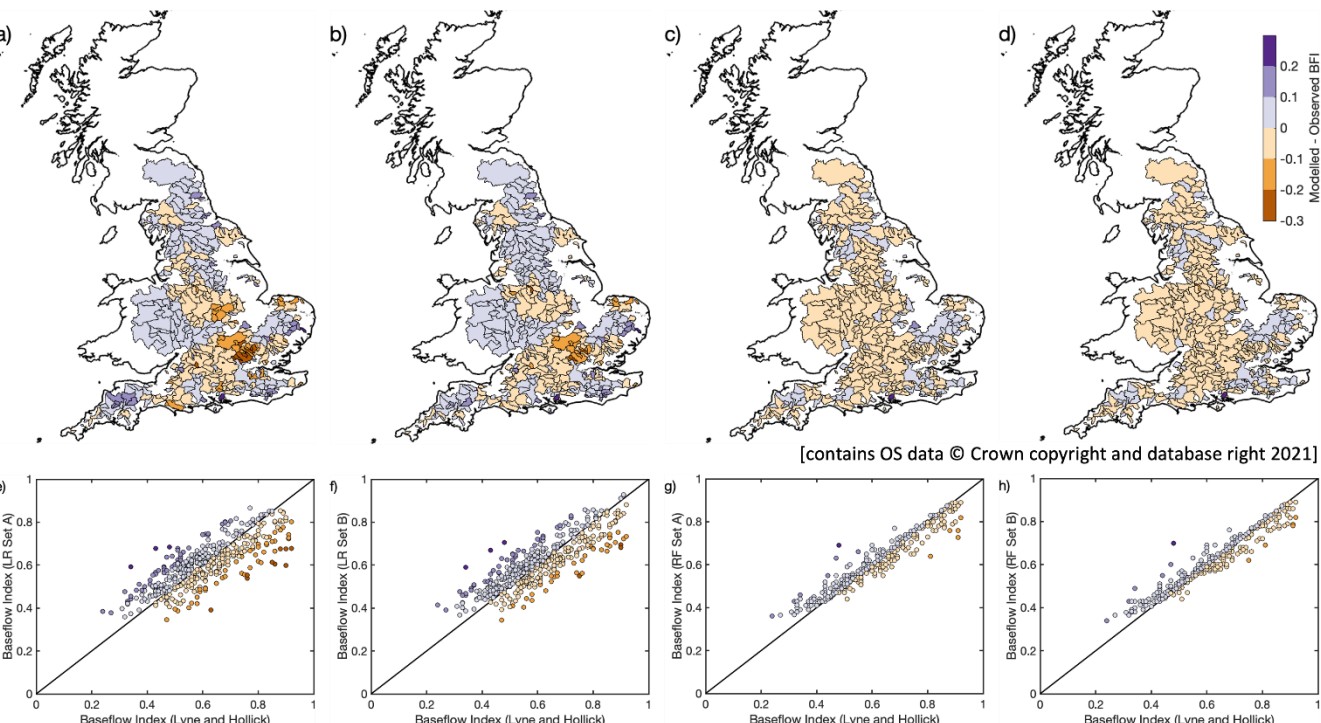

**Figure 6.** Maps of difference between modelled and observed BFI_LH (a to d) and corresponding scatter plots of BFI_LH against fitted BFI (e to h) for covariate Sets A and B for LR and RF models (Models 1, 5, 3 and 7 respectively)

To explore further which WRM covariates (groundwater abstraction, surface water abstraction, and effluent discharges) may be contributing to the improvement of the LR models, the distribution of differences between model estimates and observed BFI as a function of the magnitude of the three WRM covariates have been plotted for BFI_LH (Fig. 8) and for BFI_CEH (Fig. A4). Figure 8 shows that for LR models using natural covariates Set A (Model 1), underestimation of BFI is greater in catchments with higher levels of groundwater abstraction and, to a lesser extent, with higher effluent discharges.

Whereas, there is no apparent systematic association between under- or overestimation of BFI_LH and levels of surface water abstraction. When the WRM covariates are included in the models (Set B, Model 5), estimates of BFI_LH are noticeably improved in catchments with high levels of groundwater abstraction and to a lesser extent moderate to high effluent discharges. Similar patterns are seen for models of BFI_CEH (Fig. A4). From this it is inferred that most of the improvement in the LR model performance when WRM covariates are included in the models is due to the groundwater abstraction covariate and, to

a lesser extent, to the discharge covariate. Inclusion of the surface water abstraction covariate appears to have a negligible influence on estimates of BFI using LR models.

           Compared with the LR models, differences between estimates of BFI from the RF models and observed values of BFI_LH and BFI_CEH using Set A covariates (Models 3 and 4) are small and there are no clear regional patterns in model performance across the study area (Fig. 6 and A2). Figure 8 shows that RF models of BFI_LH using Set A (Model 3) covariates

underestimate BFI in catchments with the highest levels of groundwater abstraction but there is no clear association between the performance of these models and levels of surface water abstraction or effluent discharges. Inclusion of WRM covariates in the RF model of BFI_LH (Set B, Model 7) does not appear to improve the model (Fig. 7 and A3) or change these relationships: BFI is still underestimated in catchments with the highest levels of groundwater abstraction and there is still no clear association between model performance and levels of surface water abstraction or effluent discharges. Similar

relationships also hold for the RF models of BFI_CEH (Fig. A4). There is no noticeable improvement in the performance of the RF models with the inclusion of WRM covariates.

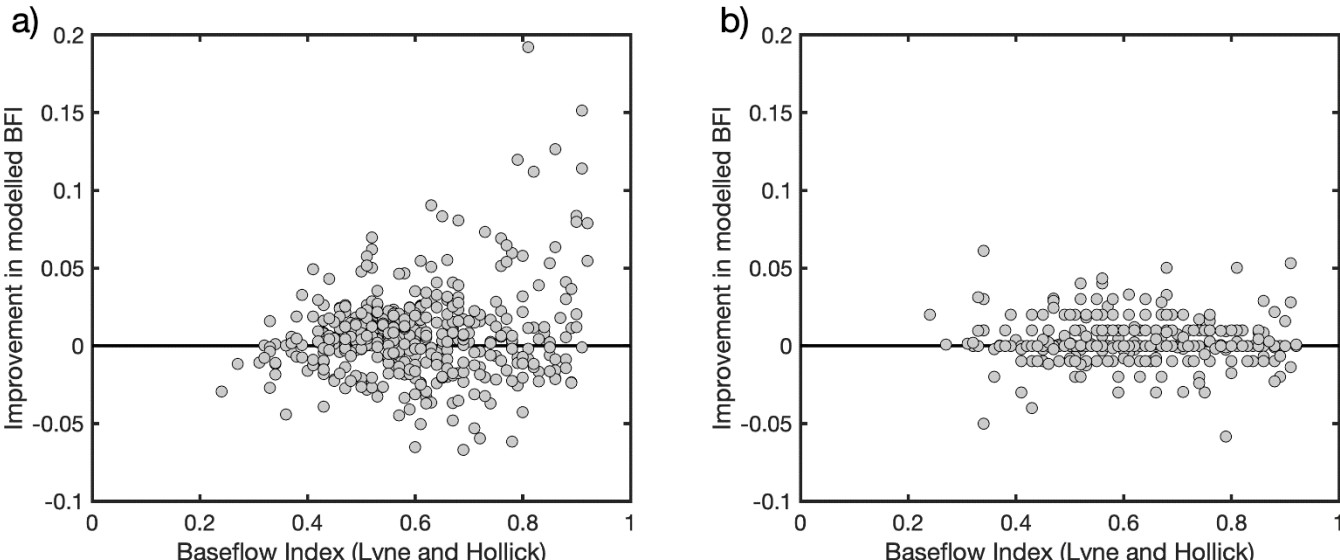

**Figure 7.** Scatter plots of improvement in modelled BFI as a function of observed BFI_LH for a) LR and b) RF models


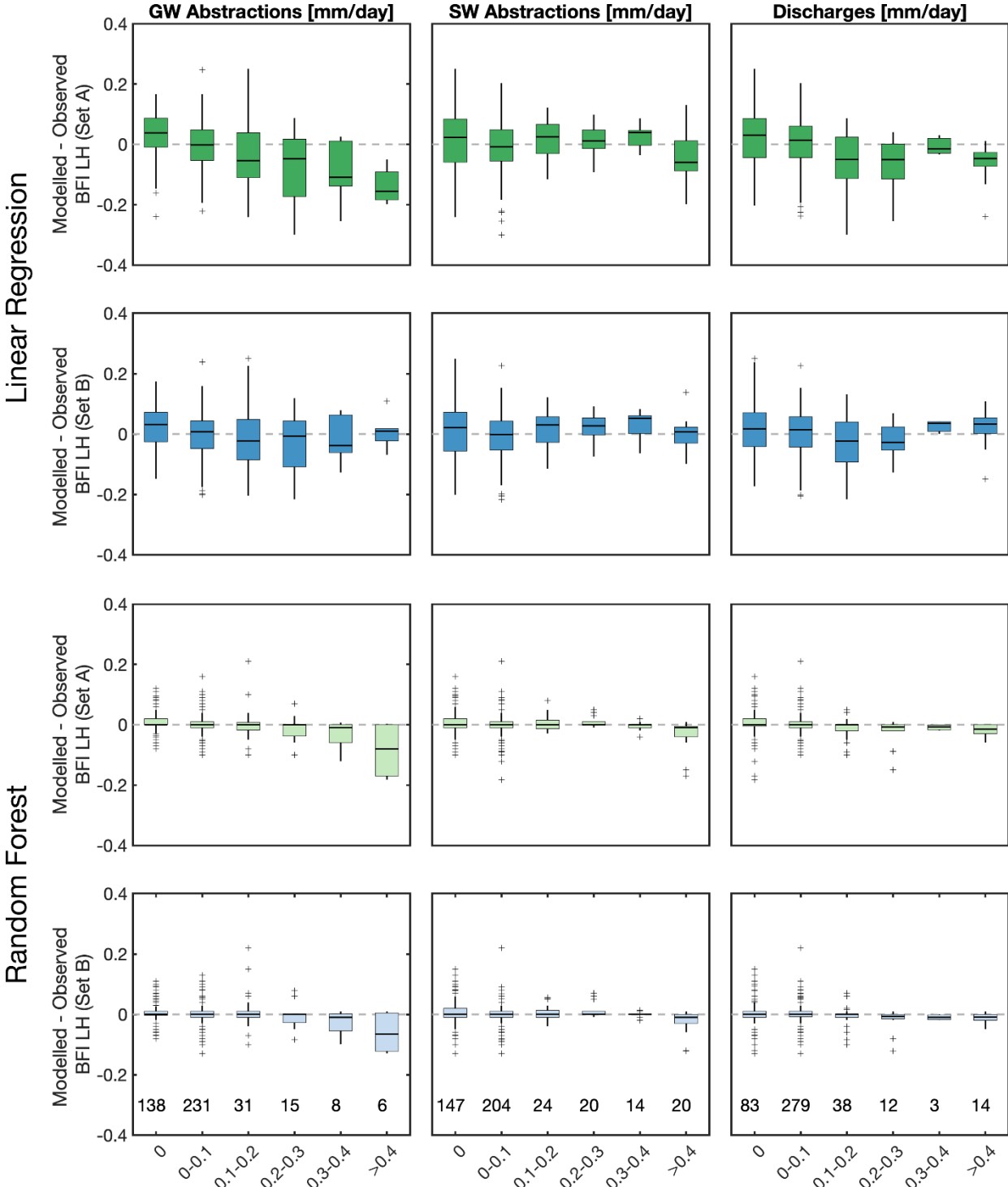

**Figure 8.** Comparison of observed and modelled BFI_LH for covariate Sets A and B, for LR and RF models and as a function of different human management categories.

## 5. Discussion

### 5.1 Impacts of WRM practices on BFI

Both modelling approaches are broadly consistent in identifying the most influential WRM covariates, namely: the importance of groundwater abstraction; the modest effect of effluent discharges to streams; and, the unimportance of reservoirs in influencing BFI. While surface water abstraction was identified as significant in the LR model but unimportant in the RF model (Fig. 5). In addition, the LR models identified positive correlations between BFI and groundwater abstraction, surface water abstraction and effluent discharges (Fig. 5), and the influence of groundwater abstraction on BFI increases with increased abstraction (Figs. 7 and 8). It is evident from previous studies (Wittenburg, 2003; Webber and Perry, 2006; Wang and Cai, 2009, Thomas et al. 2013) that there is no universal relationship between WRM practices and baseflow, and the influence of WRM practices on baseflow is sensitive to climate, the location of abstraction in a catchment and on the details of abstraction and that in the context of the present study, the relationship between WRM practices and BFI is only partly explained in terms of the conceptual model in Fig. 4.

Assuming the principal uses for abstracted groundwater in the UK are for public supply (UK Government, 2020) where losses to evaporation are limited, abstracted groundwater from up-catchment sites should have a broadly neutral effect on baseflow. In contrast, groundwater abstracted from down-catchment or in the immediate vicinity of streams may be expected to reduce baseflow. However, neither of these simple conceptualisations of groundwater abstraction explain the positive correlation between groundwater abstraction and increased baseflow in the CAMELS-GB data (Figs. 5, 7 and 8). Water resources in England have been well-regulated within the context of the European Water Framework Directive and daughter Directives (European Commission, 2000), and a wide range of sophisticated schemes and measures are used to manage low flow and drought including: conjunctive use schemes, low flow alleviation schemes, and hands-off flow measures (Clayton et al., 2008; Shepley et al., 2009; Agnew et al., 2000; Hutchinson et al., 2012; Wendt et al., 2021). Conjunctive use schemes use combined management of groundwater and surface water abstractions to maintain ecological flows while low flow alleviation schemes and hands-off flow measures are used in England to constrain the amount of water that is abstracted from groundwater and rivers, with abstractions being reduced or stopped at a given low flow trigger levels. Unfortunately, the CAMELS-GB data does not capture the details of any of these schemes or measures, and the conceptualisation of baseflow generation in Fig. 4 dose not capture the temporally and spatially linked changes in flows associated with these schemes and measures. In addition, although the analysis presented here uses BFI data for the period 1970 to 2015, the schemes and measures have evolved significantly over this period and so are both temporally and spatially variable. Consequently, although the cumulative, spatio-temporally varying effects of these schemes and measures may influence the relationship between WRM terms in the models, because there is no information on the dynamic management of water resources in the CAMELS-GB data in response to hydro-meteorological events (beyond the average terms used in the study, Table A1) the effects of the schemes

and measures on BFI cannot be constrained by the present study. The positive correlation between effluent discharges and BFI is consistent with the conceptualisation of baseflow generation in Fig. 4 while the lack of any significant or important correlation between the terms associated with reservoirs and BFI (Fig. 5) is consistent with the conceptualisation of these as stores of water that do not contribute to baseflow (Fig. 4).

## 5.2 Impacts of climate and landscape characteristics on BFI

Both modelling approaches point to the same natural covariates (Models 1 to 4) contributing to the majority of variation in BFI (Figure 5). These include a climate covariate (aridity), a number of catchment characteristics including topography (catchment mean drainage path slope, dpsbar), fractional area of highly productive fractured aquifer (frac_high_perc), non-aquifer (no_gw), and the clay fraction in soils (clay_perc), and a land cover characteristic (fractional area of crop cover, crop_perc). Qualitatively there is consistency between these covariates and similar covariates identified in previous studies. For example, Mazvimavi et al (2005) also found slope to be a significant term in a regression model of BFI for 52 basins in Zimbabwe, and Addor et al (2018) found slope to be an important covariate in an analysis of the CAMELS data for the USA. Note the observation in Table A1 that when topographic relief appears to be more important with respect to mean residence and transit times, catchment area appears less important. This is consistent with the results in both Fig. 5 and Addor et al., (2018). Beck et al (2013) demonstrated that PET (a climate covariate related to aridity) was a significant covariate in a regression model of BFI based on 3394 global catchments consistent with the results in Fig. 5. Bloomfield et al (2009) previously identified the importance of the fractional area of high productivity fractured aquifers and non-aquifers in controlling BFI in the Thames Basin, a basin within the current study area, again consistent with the results in Fig. 5. Similarly, Addor et al (2018) and Huang et al (2021) both found clay fraction in soils to be important in predicting BFI when ML techniques were applied to the CAMELS data for the USA.

However, there are challenges in making direct comparisons between different models of BFI. Firstly, there is no commonly accepted approach to defining covariates used in such models. Although many of the climate and topographic catchment characteristics may have common definitions, other important or significant catchment factors, such soil and aquifer characteristics may be quantified quite differently between studies. The CAMELS family of hydrological large-sample datasets seek to address the issue of consistency between hydrological datasets by attempting to published hydrological data in standardised formats (Addor et al., 2020). However, even between the different national CAMELS datasets there are differences in how (hydro)geological attributes are characterised (Addor et al., 2017; 2020; Alvarez-Garreton et al., 2018; Chagas et al., 2020; Coxon et al., 2020a, 2020b). A second challenge when attempting to compare between studies of the natural controls on BFI is that studies typically investigate different combinations of covariates. Regardless of the modelling approach used, for example step-wise multiple LR (e.g. Mazvimavi et al., 2005; Bloomfield et al., 2009; Zhang et al., 2013; Aboelnour et al., 2021) or ML models (Mazvimavi et al., 2005; Addor et al., 2018; Huang et al., 2021), the resulting significant or important covariates reflect the composition of the original pool of covariates under consideration.

### 5.3 Implications for future research

There are a couple of implications that arise from this study. Although the dominant controls on baseflow across the study area are climate and catchment covariates, there is evidence that WRM practices, particularly groundwater abstraction, influence baseflow but the manner in which they effect baseflow is inferred to be a function of the specific climate and catchment settings and WRM practices. Consequently, as this analysis and the CAMELS-GB data reflect the dominant WRM practices for GB, it is recommended that the present study should be extended to test additional WRM attributes and the

applicability of the findings in other settings and WRM regimes. For example, CAMELS-GB does not explicitly include information about WRM practices associated with hydropower schemes or seasonal changes in abstraction (e.g. for irrigation), so the effect of such WRM practices on BFI has not been assessed. In addition, CAMELS-GB does not include any information on within and between catchment water transfers (note the absence of these WRM terms from the conceptual model, Fig. 4). In addition, the approach to assessing the effect of WRM practices on BFI could also be applied and tested for relevance in

other climate settings such semi-arid environments (Mwakalila et al., 2002), or where snowmelt is an important component of baseflow generation (Miller et al., 2014; Barnhart et al., 2016; Huang et al., 2021) once systematic information on WRM practices is available in those settings.

        More broadly, it is important to make data related to WRM practices much more widely available and for that data to be included in future large-catchment datasets (Addor et al., 2020). It is already challenging to develop common approaches

to characterise some important catchment covariates related to soils and (hydro)geology for inclusion in large-catchment datasets. It is likely to be even more difficult to provide a consistent approach to capturing WRM practice data. However, a starting point would be to systematically conceptualise the major WRM practices across a wide range of regulatory (unregulated to highly regulated), catchment, and climatic settings that may influence baseflow and other hydrological signatures (McMillan, 2021) in order to establish broad classes of WRM practices against which data can be reported.

Finally, there is an active debate on the comparative merits of process-based hydrological modelling and ML in hydrological forecasting. Specifically, questions have been asked related to the extent to which hydrological processes and our understanding of the uniqueness of place, as encapsulated in our conceptual models of the terrestrial water cycle (Wagener et al., 2021), has a role in hydrological prediction in the 'age of machine learning' (Bevan 2020; Nearing et al 2020). For example, in a recent comparative study of the predictive accuracy of ML and LR models of flooding events in Germany, Schmidt et al.,

(2020) demonstrated that although ML methods had higher predictive accuracy than the LR models they were still shown to be susceptible to the problem of equifinality and that this severely restricted their potential for inference. Schmidt et al., (2020) concluded with the observation that multiple algorithms and multiple methods should ideally be employed within a framework of model cross-validation when using ML for inference. Although the purpose of the present modelling was not to develop models capable of predicting BFI, it is interesting to note that there have been clear benefits in applying both simple statistical

models (LR models) and more flexible ML approaches (RF models) to the same parameter space to explore common model structures and covariates of interest, and the results have provided evidence to extend current process understanding of

baseflow based beyond individual LR (Bloomfield at al., 2009; Carlier et al., 2018; Zhang et al 2020) and RF (Mazvimavi et al., 2005; Addor, et al., 2018; Huang et al., 2021) studies. Now that the correlations between WRM covariates and BFI have been identified, future predictive models of BFI that take account of WRM practices could be developed using a refinement of the conceptual model (Fig. 4) to constrain a combination of multiple targeted statistical (LR) and multiple knowledge-guided ML models (Shen et al., 2021) deployed with appropriate cross-validation schemes.

## 6. Conclusions

Variation in BFI is predominantly explained by natural (climatic and catchment) characteristics, with the most important being the extent of high productivity fractured aquifers within catchments. This latter observation being consistent with previous analyses of BFI within the study area. Although not the major control on variation in BFI, there is evidence that WRM practices systematically modify BFI in the study area.

Groundwater abstraction is the most influential of these practices with a positive correlation between abstraction and baseflow and is consistent with the observation that the effect of groundwater abstraction on BFI is most evident in groundwater-dominated catchments where there are the highest levels of abstraction. However, a variety of schemes and measures are used to manage water resources in the UK and systematic information on such schemes is currently lacking in the CAMELS-GB large sample dataset so their specific effects on BFI cannot be constrained by the current study. Information regarding WRM practices, their temporally and spatially linked associations and changes in flows associated with these schemes and measures, should be incorporated in future conceptual models of BFI.

Large-sample datasets are increasingly being used to understand and predict the functioning of hydrological systems at scales above the individual catchment (Addor et al., 2020). Given the importance of understanding the effects of WRM practices on baseflow and a range of other hydrological signatures there is need to incorporate information about such practices in large-sample datasets. If such datasets are to be comparable, there is also the need to systematise how WRM practices, in all their diversity, are described and recorded.

## 7. Data availability

The CAMELS-GB dataset used in this study is available from the Environmental Information Data Centre (EIDC) at https://doi.org/10.5285/8344e4f3-d2ea-44f5-8afa-86d2987543a9 and the supporting paper describing the data (Coxon et al., 2020b) is available from Earth System Science Data at https://essd.copernicus.org/articles/12/2459/2020/

## 8. Author contributions

JPB designed the study and undertook the literature review. MG and BPM performed the modelling and all authors contributed to the analysis of the results. JPB prepared the manuscript and GC prepared the figures (except Fig. 4 and 5 which were produced by JPB). GC, MG, BPM and NA reviewed and edited the manuscript.

## 9. Competing interests

The authors declare that they have no conflict of interest.

## 10. Acknowledgements

JPB and BPM publish with the permission of the Director, British Geological Survey (Natural Environment Research Council, UKRI).

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

## 12. Appendix

**Table A1.** Description of the CAMESL-GB covariates used in the modelling and analysis.

| Covariate class | CAMELS-GB covariate | Details of CAMELS-GB covariate | Context |
|---|---|---|---|
| *Catchment physiography* | area | Catchment area (km$^2$) based on date from UKCEH's Integrated Hydrological DTM (Morris and Flavin, 1990). | Catchment area is commonly identified as an important factor in explaining variability in low flows (Price et al., 2011). However, it less important with respect to mean residence and transit times where topographic relief appears to be more important (McGlynn et al, 2003; Asano and Uchida, 2012; Munoz-Villiers et al., 2016). |
| | dpsbar | Catchment mean drainage slope path (m km$^{-1}$). | Mean drainage path slope (Bayliss, 1999) is an index of catchment |

| | | | steepness and is estimated as the mean of all inter-nodal slopes from UKCEH's Integrated Hydrological DTM for a given catchment (Morris and Flavin, 1990). |
|---|---|---|---|
| *Climate indices* | aridity | Aridity (-). Aridity in CAMELS-GB, as with the other CAMELS data sets, is calculated as the ratio of mean daily potential evapotranspiration to mean daily precipitation (Addor et al., 2017, Coxon et al., 2020b). In the present study it has been reformulated as usually estimated (Joint Research Centre, 2019). | The primary input to the catchment water balance and hence to baseflow generation is precipitation minus evapotranspiration (Price 2011, Fig.1). |
| | frac_snow | Fraction of precipitation falling as snow (for days colder than 0°C) was estimated by Coxon et al., (2020b). | Barnhart et al (2016) demonstrated a strong correlation between snowmelt rate and baseflow efficiency for catchments from western USA. |
| *Hydrogeology classes* | inter_high_perc | Percentage of catchment designated as being underlain by rock with intergranular flow & high productivity (%) (Hydrogeological attributes for each catchment were derived from the UK bedrock hydrogeological maps, British Geological Survey, 2019). | As Price (2011) notes, catchment geology is a primary control on baseflow-generating process. Three of the nine CAMELS-GB hydrogeological attributes have been selected as covariates, these include the two high groundwater productivity attributes and the attribute that denotes essentially no groundwater. Bloomfield et al., (2009) had previously explained 97% of the variance in BFI for 44 catchments in the Thames Basin, UK, using a model that regressed four hydrogeological classes including two high productivity and two low productivity classes on BFI. |
| | fract_high_perc | Percentage of catchment designated as being underlain by rock with flow through fractures & high productivity (%). | |
| | no_gw_perc | Percentage of catchment designated as being underlain by rocks with essentially no groundwater (%). | |
| *Land cover* | dwood_perc | Percentage of catchment designated as deciduous woodland coverage (%) (Attributes for each catchment were derived from the UK Land Cover Map | Processes associated with the transformation of the hydrological inputs, in forests and shrubby vegetation, such as interception, |

| | | | |
|---|---|---|---|
| | | 2015 produced by UKCEH, Rowland et al., 2017). | throughflow and stem flow, at the ground surface, such as ponding and infiltration, and in the soil, such as deep drainage and recharge (Price, 2011) depend on the nature of land use and land cover. |
| | ewood_perc | Percentage of catchment designated as evergreen woodland coverage (%). | |
| | grass_perc | Percentage of catchment designated as grass and pasture coverage (%). | |
| | shrub_perc | Percentage of catchment designated as medium scale vegetation (shrubs) coverage (%). | |
| | crop_perc | Percentage of catchment designated as crops coverage (%). | |
| | urban_perc | Percentage of catchment designated as suburban and urban coverage (%). | |
| | interwater_perc | Percentage of catchment designated as inland water coverage (%). | |
| | bares_perc | Percentage of catchment designated as bare soil and rocks coverage (%). | |
| *Soil* | clay_perc | Percentage clay content of soil (%). Soil attributes for each catchment were based on the European Soil Database Derived Data product (Hiederer, 2013). | Using data from over 600 catchments in the CAMELS-US dataset, Addor et al., (2018) used ML to compare the influence of catchment attributes on a variety of hydraulic signatures including BFI_LH. Soil clay fraction was the most negatively correlated attribute with BFI_LH (Addor et al., 2012, Fig. 4). |
| *Water resource management* | surfacewater_abs | Mean surface water abstraction (mm day$^{-1}$). Mean surface water and groundwater abstraction and discharge data were estimated by Coxon et al., (2020) based on monthly actual abstractions and returns for the period January 1999 – December 2014. | Wittenburg (2003), Wang and Cai (2009), Webber and Perry (2006) and Tomas et al. (2013) have all previously identified changes in features of baseflow in catchments subject to groundwater abstraction or due to returns flows. |
| | groundwater_abs | Mean groundwater abstraction (mm day$^{-1}$). | |

| | | |
|---|---|---|
| discharges | Mean discharges (mm day$^{-1}$). Discharge data consists of daily discharges into water courses from water companies and other discharge permit holders reported to the Environment Agency from 1 January 2005 to 31 December 2015. | |
| num_reservoirs | Number of reservoirs in the Catchment (-). Reservoir attributes were taken from an open source UK reservoir inventory (Durant and Counsell, 2018). | |
| reservoir_cap | Total storage capacity of reservoirs in the catchment (ML). | |


**Table A2.** Coefficients of the four LR models, and associated spatial structural parameters and summary statistics for the models

| | Model 1 Set A BFI_LH | Model 2 Set B BFI_CEH | Model 3 Set A BFI_LH | Model 4 Set B BFI_CEH |
|---|---|---|---|---|
| intercept | 1.3652 | 1.4068 | 1.1372 | 1.2137 |
| dpsbar | 0.0029** | 0.0056*** | 0.0034*** | 0.0063*** |
| aridity | -0.2182*** | -0.3220*** | 0.238*** | -0.4002 |
| inter_high_perc | | | | (-0.0031***) |
| frac_high_perc | 0.0107*** | 0.0194*** | 0.0105*** | 0.0031*** |
| no_gw_perc | -0.0028** | -0.0035* | -0.0018* | -0.0021*** |
| crop_perc | 0.0096*** | 0.0157*** | 0.0089*** | 0.0149*** |
| urban_perc | (0.0027***) | (0.0032***) | | |
| inwater_perc | (0.0850***) | | | |
| clay_perc | -0.0412*** | -0.0538*** | -0.0350*** | -0.0476*** |
| surfacewater_abs | | | 0.3278*** | 0.5239*** |
| groundwater_abs | | | 1.3861*** | 1.8737*** |
| discharges | | | 0.7099*** | 0.7285*** |
| Spatial structure parameters | | | | |

| | | | | |
|---|---|---|---|---|
| Range | 0.504 | 0.446 | 0.473 | 0.426 |
| Nugget | 0.408 | 0.383 | 0.496 | 0.387 |
| Summary of models | | | | |
| MSPE | 0.117 | 0.360 | 0.138 | 0.289 |
| Residual std. | 0.435 | 0.642 | 0.388 | 0.581 |
| R^2 (iid model) | 0.627 | 0.669 | 0.703 | 0.732 |


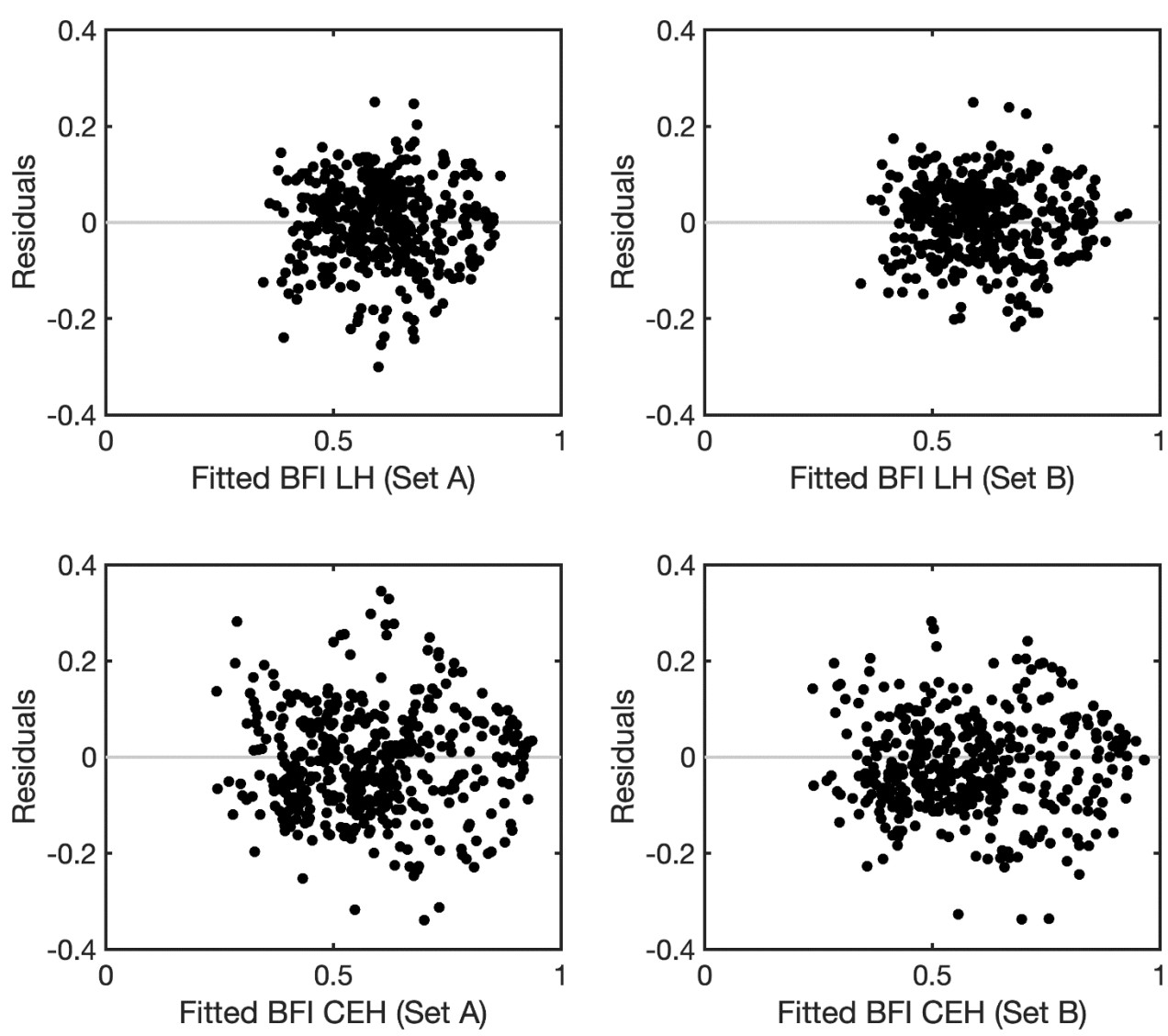

**Figure A1.** Distribution of residuals for LR models (Models 1 to 4)

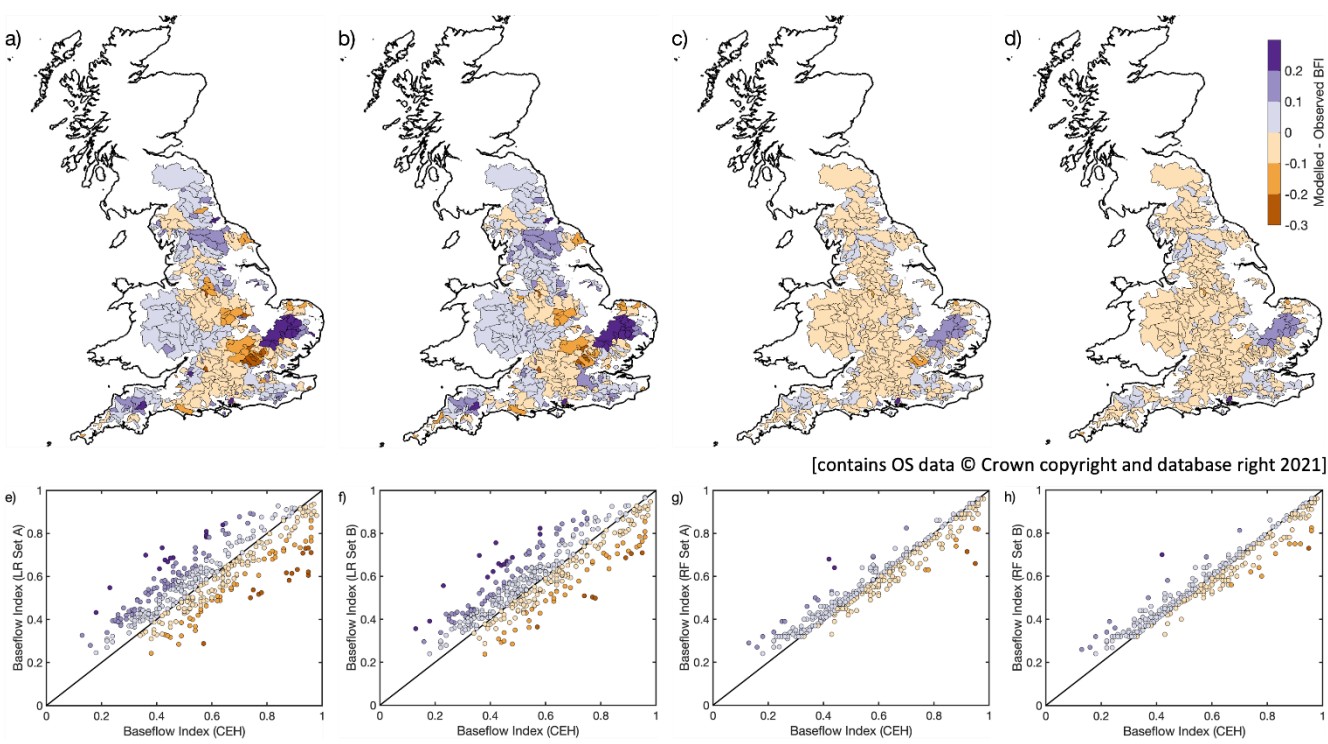

[contains OS data © Crown copyright and database right 2021]

**Figure A2**. Maps of difference between modelled and observed BFI_CEH (a to d) and corresponding scatter plots of BFI_CEH against modelled BFI (e to h) for covariate Sets A and B for LR and RF models (Models 2, 4, 6 and 8) [Note this is the same as Fig. 6, but for BFI_CEH].

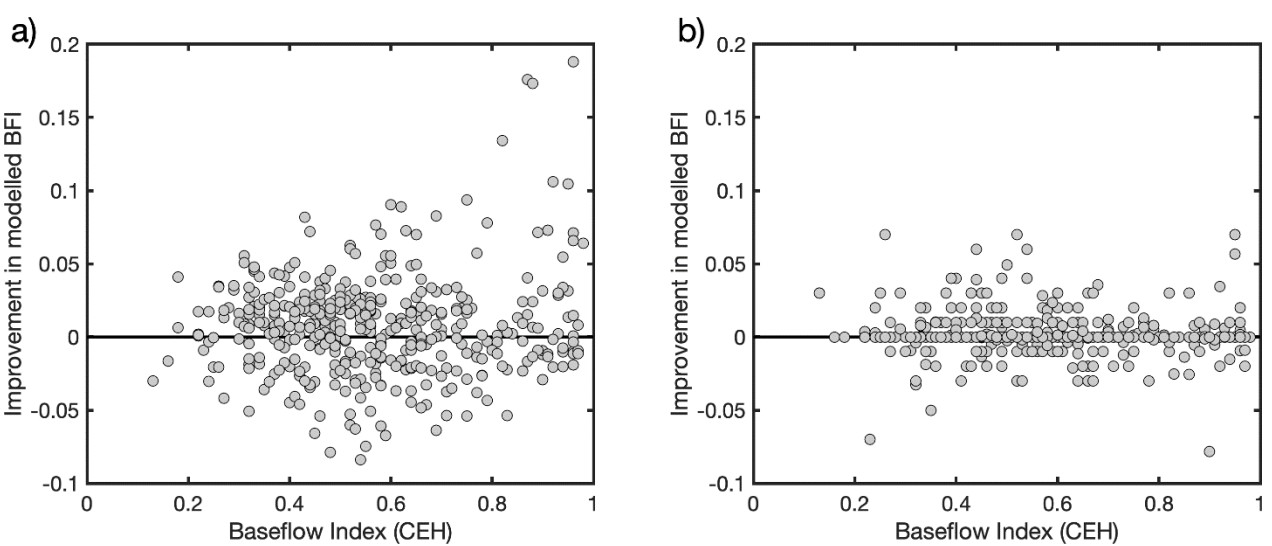

**Figure A3.** Scatter plots of improvement in modelled BFI as a function of observed BFI_CEH for a) LR and b) RF models. [Note this is the same as Fig. 7, but for BFI_CEH].

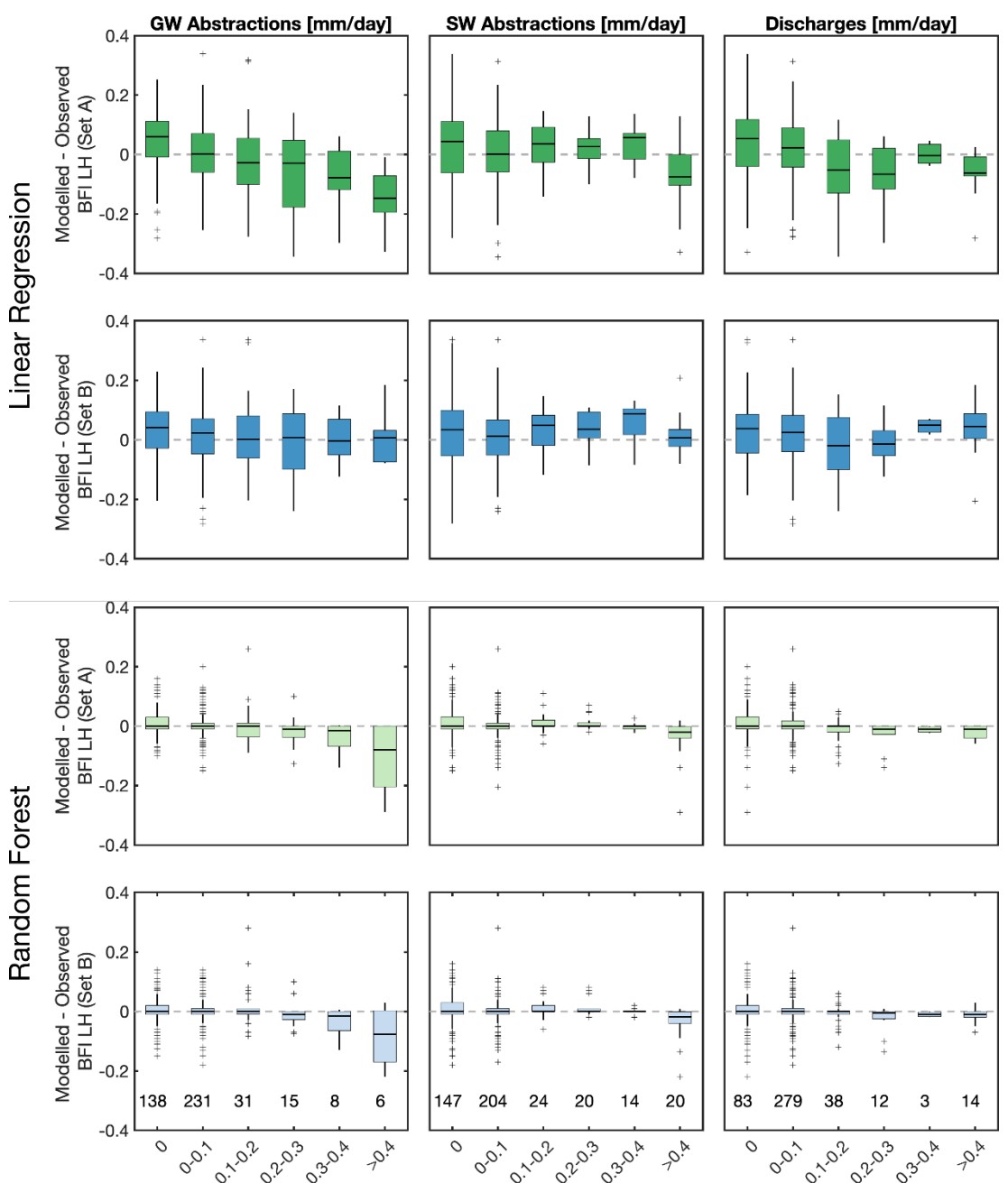

**Figure A4.** Comparison of observed and modelled BFI_CEH for covariate Sets A and B, for LR and RF models and as a function of different human management categories. [Note this is the same as Fig. 8, but for BFI_CEH].