# Peer review of "How is Baseflow Index (BFI) impacted by water resource management practices?"

_Hydrology and Earth System Sciences, 2021_

## Author Comment (AC1)

**Response to reviewer #1**

We would like to thank Anonymous Reviewer # 1 for their comments.

**Comment 1**

With respect to the first part of comment 1, information needed to understand the water resource management (WRM) practices that are the focus of the present study has been set out in detail in the paper. The source of the data used to analyse the WRM practices and conceptually how those practices may be related to BFI have been described. We also note, as part of the Discussion, which WRM practices are out of scope with respect to the current study.

In this context, the following sections of the paper are particularly pertinent:

- In the Introduction (lines 44-57) we cite and describe studies that have previously investigated how WRM practices may affect baseflow.
- In the Introduction (Lines 62-65) we describe which WRM practices are the focus for the study and link these to specific aims of the study, as follows:
  > "CAMELS-GB, a recently published large-sample hydrology dataset for Great Britain (GB) (Coxon et al., 2020a; 2020b), is unusual in that it contains quantitative information on WRM practices including surface water and groundwater abstractions, discharges, and reservoir numbers and capacities at the catchment scale. The aim of the present study is to use the CAMELS-GB large-sample dataset to identify which, if any, WRM activities influence baseflow; to assess the importance of these activities in the context of other factors known to influence baseflow, such as meteorology, catchment hydrogeology, catchment physiography, and LCLU; and, to investigate if WRM factors are important in any particular catchment or management settings.".
- In section 2.2 we note that the study is using WRM data taken from Coxon et al., (2020a; 2020b) where the nature of the management practices and source of the data are described in full.
- In section 2.2 at lines 148-156 (including Figure 4), we show how the WRM practices are conceptualised in the present study based on a modification of a conceptual model of baseflow generation after Price et al., (2011).
- In the Discussion at line 448-457 we describe WRM practices that are not in scope based on the present CAMELS-GB data and catchment settings and briefly describe what additional work could be undertaken to address this issue.

The second part of comment 1 raises the question of uncertainty in the estimates of BFI. BFI is a hydrological signature (McMillan, 2021) that can be estimated using a wide range of techniques from a wide range of data sources. To account for the uncertainty in calculating BFI, we have used two techniques (Lyne-Hollick and CEH method). Although there are small differences in the BFI estimates the conclusions that we draw between these techniques are consistent. In terms of uncertainties in the underlying data, we recognise that there are often large uncertainties in streamflow data (Coxon et al, 2015) but these are difficult to characterise across large samples of catchments and uncertainty estimates are not available for all the CAMELS-GB catchments. We also note that BFI typically has lower uncertainty compared with other hydrological signatures as it is based on temporal averaging (Westerberg and McMillan, 2015). We agree that this is an important point to consider and a note on uncertainty related to BFI estimates based on the above comments will be added to section 2.2 of a revised version of this paper.

**Comment 2**

The "practical implication" of the study is summarised in the final point of the conclusions, namely that:

> "WRM practices can and should where appropriate be incorporated in future conceptual models of BFI and baseflow generation, and consequently data and information about WRM practices should be included in future large-sample catchment datasets and in future investigations of baseflow".

However, we acknowledge that this could be expanded on in the Discussion and we will add a short note to that effect in the revised paper.

**Comment 3**

We will work with HESS to improve the picture quality in a revised draft.

**Comment 4**

There is no repetition. However for clarity the text will be revised to read as follows:

> "Figure 3c shows that that there is a generally good linear agreement between the two estimated BFI indices. However, for BFIs below 0.7 BFI_CEH is systematically lower than BFI_LH and for BFIs above 0.7 BFI_CEH is systematically higher than BFI_LH."

**Comment 5**

In the revised manuscript we will add the equation for Lin's concordance coefficient:

$$\rho_c(x,y) = \frac{2\rho(x,y)\sqrt{\mathrm{var}(x)}\sqrt{\mathrm{var}(y)}}{\mathrm{var}(x) + \mathrm{var}(y) + \left(\mu_x - \mu_y\right)^2},$$

where $\rho_c(x,y)$ is Lin's concordance coefficient for variables $x$ and $y$, $\rho(x,y)$ is Pearson's coefficient for the same variables, $\mathrm{var}(x)$ is the variance of $x$ and $\mu_x$ in the mean of $x$. We will expand our description to state that Lin's concordance coefficient can take values between -1 and 1, that a value of 1 indicates an exact match between the two variables and that the $\left(\mu_x - \mu_y\right)^2$ term means that variables with different mean values have a small coefficient value in contrast to standard correlation coefficients where perfectly correlated variables can have vastly different mean values.

**Comment 6**

A description of the modelling scheme and nomenclature is provided at the start of modelling methods, section 3 (Lines 157-166). However, for clarity a brief note related to the modelling scheme nomenclature will be added at the start of the results section, section 4, where Figure 5 is introduced.

**Comment 7**

Catchment area was included in the covariates for analysis since, as we note in Table A1 (Line 700) "catchment area is commonly identified as an important factor in explaining variability in low flows (Price et al., 2011)". However, as the reviewer notes, catchment area was not identified as significant or important in either the LR or RF models respectively (Figure 5). It is not certain why this is the case. However, we also noted at Line 700 in Table A1 that catchment area is "less important with respect to mean residence and transit times where topographic relief appears to be more important" and this is consistent with the observation that, at least in the LR models (Figure 5), the

covariate 'dpsbar' (catchment mean drainage slope path (m km-1)) was significant. Other recent studies of hydrological indices, such as Addor et al., (2018) have shown that catchment area is unimportant with respect to BFI whereas slope is important (Addor et al., 2020, Figure 4).

**Comment 8**

It is not clear why the LR models are more effective than the RF models in identifying which WRM covariates (groundwater abstraction, surface water abstraction, and discharges) may be contributing to the improvement of the model performance when WRM terms are added. It may be because the RF models are overfitted. However, note that the performance of both the models is discussed in the paragraph following the commented text (e.g. Lines 412-423).

**Comment 9**

The conclusions highlight the main findings of the paper in a manner complimentary to the Abstract while hopefully avoiding the common tendency of being overly long. We would be happy to address any specific comments or suggestions related to the Conclusions

**Comment 10**

The introduction includes an overview of the literature related to baseflow and water WRM practices and puts the study in the wider context of the very large field of research related to baseflow. We would be happy to address any specific comments related to the literature cited in the paper.

**References**

Addor, N., Nearing, G., Prieto, C., Newman, A. J., Le Vine, N. and Clark, M. P.: A ranking of hydrological signatures based on their predictability in space, Water Resour. Res., 54(11), 8792–8812, 2018

Coxon, G., Freer,J., Westerberg, I.K., Wagener, T., Woods, R., and Smith, P.J.: A novel framework for discharge uncertainty quantification applied to 500 UK gauging stations, Water Resour. Res., 51, 5531–5546, 2015.

Coxon, G., Addor, N., Bloomfield, J. P., Freer, J., Fry, M., Hannaford, J., Howden, N. J. K., Lane, R., Lewis, M., Robinson, E. L., Wagener, T., Woods, R.: Catchment attributes and hydro-meteorological timeseries for 671 catchments across Great Britain (CAMELS-GB). NERC Environmental Information Data Centre, [data set], available at: https://doi.org/10.5285/8344e4f3-d2ea-44f5-8afa-86d2987543a9 (last access: 8 April 2021), 2020a.

Coxon, G., Addor, N., Bloomfield, J. P., Freer, J., Fry, M., Hannaford, J., Howden, N. J. K., Lane, R., Lewis, M., Robinson, E. L., Wagener, T., and Woods, R.: CAMELS-GB: Hydrometeorological time series and landscape attributes for 671 catchments in Great Britain, Earth Syst. Sci. Data, 12, 2459-2483, 2020b.

McMillan, H., K.: A review of hydrologic signatures and their applications, WIREs Water. 2021;8:e1499, 2021

Westerberg, I., K., and McMillan, H., K: Uncertainty in hydrological signatures, Hydrol. Earth Syst. Sci., 19, 3951–3968, 2015

---

## Author Response (AR1)

**Response to reviewers' comments**

Once again, we would like to thank both reviewers for their thoughtful and insightful comments (in bold below) which we believe, through our responses, have led to an improved manuscript. Here we present point-to-point responses to each of the comments made by both the reviewers.

**Response to Reviewer #1 (Anonymous)**

**Comment 1: "The paper does not go in depth about the different water resources management practices and it relationship with baseflow contribution. Baseflow index calculation itself have uncertainty as it was never verified with any filed data. So it would be nice to capture these uncertainty and discuss in the paper."**

Response: We agree that more detail could be provided with regard to the different WRM practices. To this end we have:

- Provided extra clarification regarding what is meant by the term WRM practices in the Introduction at lines 63 to 69, as follows:

  "Here WRM practices is a loosely defined term that encompasses a wide range of activities related to the management of groundwater and surface water resources that are specifically distinct from wider 'human influences' or 'human activities' (Zhang et al., 2019; Mo et al., 2021) that affect LULC, such as of urbanization, deforestation, and land-management practices. Wang and Kai (2009) referred to WRM practice as 'direct human interferences'. Some examples of WRM practices include abstraction and discharge, changes in conveyance of streams due to changes in channel structure for example for damming, flow regulation and flood management, and development of structures for water storage within catchments including dams and artificial wetlands."

- We have added additional details in the Introduction to the description of previous studies of the effects of the different WRM practices (see lines 70 to 97) including a concluding observation that:

  "In summary, as with natural controls on baseflow (Gnann et al., 2019), there is as yet no general theory to explain the effects of WRM practices on baseflow, and the effect of a given WRM practice on baseflow may be contingent on a range of factors including climate, (hydro)geological setting, location, and timing of the activity."

- We have clarified how the data used in the current study relates to the conceptualisation of WRM processes in section 2.2 (lines 199 to 203 and 208 to 2011), as follows:

  "Five WRM covariates from the CAMELS-GB dataset have been selected for analysis: groundwater abstraction (groundwater_abs), surface water abstraction (surfacewater_abs), effluent discharges (discharges) to streams and the number and capacity of reservoirs within catchments (num_reservoirs and reservoir_cap). Note that the discharge term only accounts for effluent from sewage treatment works and does not provide information on other water returns (Coxon et al., 2020b)."

  "It conceptualises WRM practices as simple high-level flows between groundwater, streamflow and components of storage. Some flows that may be significant within a given catchment, such as mains leakage (conceptualised Fig. 4), however these are

outside the current analysis as there is no information for these flows in CAMELS-GB."

- In the Discussion we have extended the text to better describe the implications of WRM practices specific to the study area for baseflow. The new text is at lines 496 to 591, as follows:

"Assuming the principal uses for abstracted groundwater in the UK are for public supply (UK Government, 2020) where losses to evaporation are limited, abstracted groundwater from up-catchment sites should have a broadly neutral effect on baseflow. In contrast, groundwater abstracted from down-catchment or in the immediate vicinity of streams may be expected to reduce baseflow. However, neither of these simple conceptualisations of groundwater abstraction explain the positive correlation between groundwater abstraction and increased baseflow in the CAMELS-GB data (Figs. 5, 7 and 8). Water resources in England have been well-regulated within the context of the European Water Framework Directive and daughter Directives (European Commission, 2000), and a wide range of sophisticated schemes and measures are used to manage low flow and drought including: conjunctive use schemes, low flow alleviation schemes, and hands-off flow measures (Clayton et al., 2008; Shepley et al., 2009; Agnew et al., 2000; Hutchinson et al., 2012; Wendt et al., 2021). Conjunctive use schemes use combined management of groundwater and surface water abstractions to maintain ecological flows while low flow alleviation schemes and hands-off flow measures are used in England to constrain the amount of water that is abstracted from groundwater and rivers, with abstractions being reduced or stopped at a given low flow trigger levels. Unfortunately, the CAMELS-GB data does not capture the details of any of these schemes or measures, and the conceptualisation of baseflow generation in Fig. 4 dose not capture the temporally and spatially linked associations and changes in flows associated with these schemes and measures. In addition, although the analysis presented here uses BFI data for the period 1970 to 2015, the schemes and measures noted have evolved significantly over this period and so are both temporally and spatially variable. Consequently, although the cumulative, spatio-temporally varying effects of these schemes and measures may influence the relationship between WRM terms in the models, because there is no information on the dynamic management of water resources in the CAMELS-GB data in response to hydro-meteorological events (beyond the average terms used in the study, Table A1) the effects of the schemes and measures on BFI cannot be constrained by the present study. The positive correlation between effluent discharges and BFI is consistent with the conceptualisation of baseflow generation in Fig. 4 while the lack of any significant or important correlation between the terms associated with reservoirs and BFI (Fig. 5) is consistent with the conceptualisation of these as stores of water that do not contribute to baseflow (Fig. 4)."

Finally, the following text has been added to the Abstract:

"A wide range of schemes and measures are used to manage water resources in the UK. These include conjunctive use and low flow alleviation schemes and hands-off flow measures. Systematic information on such schemes is currently unavailable in CAMELS-GB and their specific effects on BFI cannot be constrained by the current study. Given the significance or importance of WRM terms in the models"

The second part of Comment 1 from reviewer #1 raises the question of uncertainty in the estimates of BFI. We agree that this is an important point to consider and a note on uncertainty related to BFI has been added to Section 2.2 at lines 177 to 182, as follows:

> "There are often large uncertainties in the underlying streamflow data used to estimate BFI (Coxon et al, 2015) but these are difficult to characterise across large samples of catchments and uncertainty estimates are not available for all the CAMELS-GB catchments (Coxon et al., 2020b). However, BFI typically has lower uncertainty compared with other hydrological signatures, as it is based on temporal averaging (Westerberg and McMillan, 2015), and that only typically small differences in the BFI estimates are observed in the present study based on the two methods of estimate (Fig. 3)."

**Comment 2: "Paper also lack in discussion about practical implication of the finding of the paper."**

Response: The study was not designed with specific practical outcomes in mind, rather to "The aim of the present study is to use the CAMELS-GB large-sample dataset to identify which, if any, of these WRM activities influence baseflow; to assess the importance of these activities in the context of other factors known to influence baseflow, such as meteorology, catchment hydrogeology, catchment physiography, and LULC (Price, 2011); and, to investigate if WRM factors are important in any particular catchment or management settings" (see lines 102 to 106).

However, in response to the review comment we have extended the discussion to include a new section, Section 5.3, to discuss the implications for the results of the study (see lines 549 to 586).

**Comment 3: "Figure quality is so poor hard to read the values."**

Response: Thank you for this feedback. We will work with HESS to ensure the appropriate picture quality in a revised draft.

**Comment 4: "[old] Line 135, " Figure 3c.... " there is kind of repetition of same info."**

Response: For clarity the text has been revised at lines 191 to 193 to read as follows:

> "Figure 3c shows that there is a generally good linear agreement between the two estimated BFI indices. However, for BFIs below 0.7 BFI_CEH is systematically lower than BFI_LH, and for BFIs above 0.7 BFI_CEH is systematically higher than BFI_LH. In addition, for sites above a BFI of about 0.7 the correlation between the two indices is reduced."

**Comment 5: "Paper need bit more explanation about Lin's coefficient. Please provide the equation."**

Response: The manuscript has been revised at lines 231 to 238 to read as follows:

> "where

$$\rho_c(x,y) = \frac{2\rho(x,y)\sqrt{\text{var}(x)}\sqrt{\text{var}(y)}}{\text{var}(x)+\text{var}(y)+\left(\mu_x-\mu_y\right)^2},$$

(1)

> and where $\rho_c(x,y)$ is Lin's concordance coefficient for variables $x$ and $y$, $\rho(x,y)$ is Pearson's coefficient for the same variables, $\text{var}(x)$ is the variance of $x$ and $\mu_x$ in the mean of $x$. Lin's concordance coefficient can take values between -1 and 1. A value of 1 indicates an exact match between the two variables and the $\left(\mu_x-\mu_y\right)^2$ term means that variables with

different mean values have a small coefficient value in contrast to standard correlation coefficients where perfectly correlated variables can have vastly different mean values."

**Comment 6: "Somewhere in text it required to give details about different model, e.g. Model 1, Model 2 in Figure 5."**

Response: A description of the modelling scheme is provided at the start of Modelling methods, Section 3. However, we agree with Reviewer #1 that the model nomenclature could be made more explicit so we have modified the text in Section 3 at lines 224 to 226 to read as follows:

> "Consequently, eight models (Models 1 to 8) have been developed and evaluated. The LR and RF models are first calibrated for the Set A covariates (Models 1 to 4), then a second separate calibration is undertaken using Set B covariates (Models 5 to 8)."

and for clarity we have also amended the text throughout wherever it is appropriate to specify a particular model run. Note, we have also amended the headers to Table 1 at line 375 as there was an error in reference to the model runs.

**Comment 7: "I was very surprised Area of the catchment is not that import for the BFI? any comments"**

Response: Catchment area was included in the covariates for analysis since, as we note in Table A1 "catchment area is commonly identified as an important factor in explaining variability in low flows (Price et al., 2011)". However, as the reviewer notes, catchment area was not identified as significant or important in either the LR or RF models respectively (Figure 5). However as is also noted in Table A1 catchment area is "less important with respect to mean residence and transit times where topographic relief appears to be more important" and this is consistent with the observation that, at least in the LR models (Figure 5), the covariate 'dpsbar' (catchment mean drainage slope path (m km-1)) is significant. Other recent studies of hydrological indices, such as Addor et al., (2018) have shown that catchment area is unimportant with respect to BFI whereas slope is important (Addor et al., 2020, Figure 4). To capture these observations, the following text has been added to the Discussion at lines 526 to 530:

> "For example, Mazvimavi et al (2005) also found slope to be a significant term in a regression model of BFI for 52 basins in Zimbabwe, and Addor et al (2018) found slope to be an important covariate in an analysis of the CAMELS data for the USA. Note the observation in Table A1 that when topographic relief appears to be more important with respect to mean residence and transit times, catchment area appears less important. This is consistent with the results in both Fig. 5 and Addor et al., (2018)."

**Comment 8**

**"8. [old] Line 409, " in addition ....." why ?"**

Response: Note that this introductory paragraph has been deleted as part of the restructuring of the Discussion.

**Comment 9: "Conclusion of this study is bit week, Need to rewrite."**

Response: We have redrafted the conclusions, see lines 587 to 604, as follows:

> "Variation in BFI is predominantly explained by natural (climatic and catchment) characteristics, with the most important being the extent of high productivity fractured

aquifers within catchments. This latter observation being consistent with previous analyses of BFI within the study area. Although not the major control on variation in BFI, there is evidence that WRM practices systematically modify BFI in the study area.

Groundwater abstraction is the most influential of these practices with a positive correlation between abstraction and baseflow and is consistent with the observation that the effect of groundwater abstraction on BFI is most evident in groundwater-dominated catchments where there are the highest levels of abstraction. However, a variety of schemes and measures are used to manage water resources in the UK and systematic information on such schemes is currently lacking in the CAMELS-GB large sample dataset so their specific effects on BFI cannot be constrained by the current study. Information regarding WRM practices, their temporally and spatially linked associations and changes in flows associated with these schemes and measures, should be incorporated in future conceptual models of BFI.

Large-sample datasets are increasingly being used to understand and predict the functioning of hydrological systems at scales above the individual catchment (Addor et al., 2020). Given the importance of understanding the effects of WRM practices on baseflow and a range of other hydrological signatures there is need to incorporate information about such practices in large-sample datasets. If such datasets are to be comparable, there is also the need to systematise how WRM practices, in all their diversity, are described and recorded."

**Comment 10: "Paper need some strengthening in Literature"**

Response: We have added additional citations throughout the paper in response to this comment and to Comments 1 and 2 from Reviewer #2. Notably, citations have been added to the Introduction section to provide more context related to the significance of baseflow for streamflow and the environment, on baseflow dynamics, clarification of water resource management practices being addressed by the paper, and the choice of filters, and to the Discussion.

**Response to Reviewer #2**

**Major comment 1**

**Comment: "The baseflow generation dynamics are poorly described. In particular, the authors could discuss better the choice of the filters."**

Response: We acknowledge the reviewers comment and the importance of providing a wider research context related to baseflow dynamics, consequently:

i.) the Introduction has been updated to note the importance of baseflow (lines 37 to 39), as follows

"It is important as it sustains surface flows particularly during relatively dry periods and droughts (Smathkin, 2001; Miller et al., 2016), because it supports ecological flows and ecosystem functioning (Poff et al., 1997; Boulton 2003), and is a factor in regulating streamflow quality and temperature (Jordan et al., 1997; Gomez-Velez et al., 2015; Hare et al., 2021)."

ii.) new lines 53 to 61 have been added to provide a view on the challenges related to the current absence of a general theory of baseflow generation dynamics, as follows:

"Despite this extensive work on baseflow generation dynamics, Gnann et al., (2019) observed that there is still no general theory to explain variations in baseflow between

catchments despite the strong evidence that it is largely controlled by the interaction of climate and landscape processes. They explored the role of climate in baseflow generation using baseflow data from the United States of America (USA) and the United Kingdom (UK) and found that in humid settings baseflow can be highly variable due to variations in catchment storage and wetting potential, whereas in more arid settings baseflow has much lower variability and is primarily controlled by vaporization limits. In a complementary study of 435 catchments across the contiguous US and the UK, Yao et al., (2021) found that soil water storage capacity is an important control on baseflow and that generally, BFI increases with storage capacity for a given a climate condition and decreases with aridity for a given storage capacity."

iii.) new lines 112 to 118 and 121 to 126 have been added to address the specific observation related to choice of filters, as follows:

"BFI is the ratio of baseflow volume to total flow volume expressed as a fraction (Nathan and McMahon, 1990) and can be estimated by hydrograph separation using a wide range of tracer-based and non-tracer methods (Eckhardt, 2008; Gonzales et al., 2009; Price et al., 2011). The two measures of BFI in CAMELS-GB both use non-tracer-based methods, specifically a digital filter (Lyne and Hollick, 1979) and a graphical / statistical method (Gustard et al., 1992; Piggot et al., 2005). The former, although it is not based on the physics of discharge processes, produces objective and reproducible estimates of BFI (Cheng et al., 2021), while the latter has been used previously to characterise BFI across the study area (Bloomfield, 2009)."

"Although studies of BFI typically consider multiple baseflow filters to reduce uncertainty in estimates of BFI (Chen and Teegavarapu, 2020; Kissel and Schmalz, 2020; Zhang et al., 2020), the present study is not designed either to assess the relative efficacy of the filters used to estimate BFI, nor to compare the respective efficacy of the chosen statistical models in estimating baseflow: this is not a model inter-comparison study (Refsgaard and Knudsen, 1996). Instead, the estimates of BFI and the modelling approaches are designed to provide complementary evidence for the nature and importance (or not) of WRM practices on influencing BFI based on the published CAMELS-GB data."

and, iv.) additional citations have been added throughout the Introduction to improve the context for the study (see also our response to Comment 10 from Reviewer #1).

**Major comment 2**

**Comment: The authors could compare their results with previous work aiming to relate BFI with catchment characteristics (Beck et al. (2013), for instance).**

Response: Given the aim of the present study has been to assess the effect of water resource management (WRM) practices on BFI we chose to focus the Discussion on the implications of the results with respect to WRM. However, we acknowledge that there may also be interest in the results of the Set A models (Models 1 to 4) that just considered natural covariates. We have introduced a new subsection in the discussion at lines 521 to 547 to place the results of the present study in the context of previous studies of relationships between climate and catchment characteristics and BFI, and have also included a note on the challenges in quantitative comparisons between such modelling studies as follows:

"Both modelling approaches point to the same natural covariates (Models 1 to 4) contributing to the majority of variation in BFI (Figure 5). These include a climate covariate (aridity), a number of catchment characteristics including topography (catchment mean drainage path slope, dpsbar), fractional area of highly productive fractured aquifer (frac_high_perc), non-aquifer (no_gw), and the clay fraction in soils (clay_perc), and a land cover characteristic (fractional area of crop cover, crop_perc). Qualitatively there is consistency between these covariates and similar covariates identified in previous studies. For example, Mazvimavi et al (2005) also found slope to be a significant term in a regression model of BFI for 52 basins in Zimbabwe, and Addor et al (2018) found slope to be an important covariate in an analysis of the CAMELS data for the USA. Note the observation in Table A1 that when topographic relief appears to be more important with respect to mean residence and transit times, catchment area appears less important. This is consistent with the results in both Fig. 5 and Addor et al., (2018). Beck et al (2013) demonstrated that PET (a climate covariate related to aridity) was a significant covariate in a regression model of BFI based on 3394 global catchments consistent with the results in Fig. 5. Bloomfield et al (2009) previously identified the importance of the fractional area of high productivity fractured aquifers and non-aquifers in controlling BFI in the Thames Basin, a basin within the current study area, again consistent with the results in Fig. 5. Similarly, Addor et al (2018) and Huang et al (2021) both found clay fraction in soils to be important in predicting BFI when ML techniques were applied to the CAMELS data for the USA.

However, there are challenges in making direct comparisons between different models of BFI. Firstly, there is no commonly accepted approach to defining covariates used in such models. Although many of the climate and topographic catchment characteristics may have common definitions, other important or significant catchment factors, such soil and aquifer characteristics may be quantified quite differently between studies. The CAMELS family of hydrological large-sample datasets seek to address the issue of consistency between hydrological datasets by attempting to published hydrological data in standardised formats (Addor et al., 2020). However, even between the different national CAMELS datasets there are differences in how (hydro)geological attributes are characterised (Addor et al., 2017; 2020; Alvarez-Garreton et al., 2018; Chagas et al., 2020; Coxon et al., 2020a, 2020b). A second challenge when attempting to compare between studies of the natural controls on BFI is that studies typically investigate different combinations of covariates. Regardless of the modelling approach used, for example step-wise multiple LR (e.g. Mazvimavi et al., 2005; Bloomfield et al., 2009; Zhang et al., 2013; Aboelnour et al., 2021) or ML models (Mazvimavi et al., 2005; Addor et al., 2018; Huang et al., 2021), the resulting significant or important covariates reflect the composition of the original pool of covariates under consideration)."

**Minor comment a**

**Comment: Please, avoid abbreviations like don't, as in [old] lines 145 and 146.**

Response: The text has been revised at lines 204 and 205 to remove abbreviations.

**Minor comment b**

**Comment: Ln. 177 BEI_CEH instead of BFI_CEH**

Response: The text has been amended at Line 246 to read "BFI_CEH".